# Genomics and biochemical analyses reveal a metabolon key to β-L-ODAP biosynthesis in *Lathyrus sativus*

Anne Edwards [1], Isaac Njaci[1,2,3], Abhimanyu Sarkar [1,4], Zhouqian Jiang [1,5], Gemy George Kaithakottil [6], Christopher Moore [7], Jitender Cheema[1], Clare E. M. Stevenson [1], Martin Rejzek [1], Petr Novák [8], Marielle Vigouroux [1], Martin Vickers[1], Roland H. M. Wouters [1], Pirita Paajanen [1], Burkhard Steuernagel [1], Jonathan D. Moore [1], Janet Higgins [6], David Swarbreck [6], Stefan Martens [9], Colin Y. Kim [10,11], Jing-Ke Weng [10,12], Sagadevan Mundree [3], Benjamin Kilian [13], Shiv Kumar [14], Matt Loose [7], Levi Yant [7,15], Jiří Macas [8], Trevor L. Wang [1], Cathie Martin [1] & Peter M. F. Emmrich [1,2,16] ✉

Grass pea (*Lathyrus sativus* L.) is a rich source of protein cultivated as an insurance crop in Ethiopia, Eritrea, India, Bangladesh, and Nepal. Its resilience to both drought and flooding makes it a promising crop for ensuring food security in a changing climate. The lack of genetic resources and the crop's association with the disease neurolathyrism have limited the cultivation of grass pea. Here, we present an annotated, long read-based assembly of the 6.5 Gbp *L. sativus* genome. Using this genome sequence, we have elucidated the biosynthetic pathway leading to the formation of the neurotoxin, β-L-oxalyl-2,3-diaminopropionic acid (β-L-ODAP). The final reaction of the pathway depends on an interaction between *L. sativus* acyl-activating enzyme 3 (LsAAE3) and a BAHD-acyltransferase (LsBOS) that form a metabolon activated by CoA to produce β-L-ODAP. This provides valuable insight into the best approaches for developing varieties which produce substantially less toxin.

The impacts of climate change over the course of the 21st century are driving the need for more diversified and resilient food and fodder crops, able to withstand weather extremes[1–4]. Grass pea (*Lathyrus sativus* L.) is an orphan legume crop with considerable potential for improving food security because of its tolerance to both drought and flooding[5,6]. This allows grass pea to be grown with minimal inputs[7,8] on marginal land and under adverse conditions that cause the failure of other food security crops[9–13]. Grass pea produces a neuroactive compound, β-L-oxalyl-2,3-diaminopropionic acid (β-L-ODAP), in its shoots and seeds. If grass pea is consumed in large quantities (>40% of caloric intake) for more than three months during periods of malnutrition, β-L-ODAP can cause neurolathyrism in humans, a disease marked by spastic and irreversible paralysis of the legs[14,15]. Pharmacological and

nutraceutical uses of β-L-ODAP have been proposed[16], but its role in the aetiology of neurolathyrism remains the primary limitation on more widespread use of grass pea as a food and feed. Repeated epidemics of neurolathyrism have been recorded for over 2000 years, and the disease continues to plague impoverished communities suffering malnourishment. A few varieties of grass pea with reduced β-L-ODAP content, obtained by selection from natural germplasm and classical breeding, have been released in India, Bangladesh, Australia and Ethiopia[17–22]. However, to remove the threat of neurolathyrism and fully utilise the potential of grass pea, the development of varieties with very low (well below 0.1% seed weight) or zero β-L-ODAP regardless of environmental conditions remains a key trait for improvement for ICARDA as well as countries heavily dependent on

grass pea, such as Bangladesh and Ethiopia. No adequate animal model of neurolathyrism exists and no reliable safe level of β-L-ODAP-consumption has been established, due to the complex aetiology of the disease. The use of modern approaches to crop improvement, such as genome editing or TILLING, has been hindered by the lack of genomic resources for grass pea and particularly by the lack of sequence data for genes involved in β-L-ODAP biosynthesis[23–27].

In this study, we present an annotated draft genome assembly of grass pea for the identification and selection of traits for agronomic improvement. This genomic resource will allow comparative genomic analyses between legumes, help in the development of high quality genetic and physical maps for marker-assisted and genomic selection strategies, and enable genome editing and TILLING platforms for grass pea improvement. We have used this genome assembly to characterise genes encoding enzymes in the β-L-ODAP biosynthesis pathway, offering a route to low-/zero-ODAP traits through gene knockouts. The availability of this draft genome will facilitate research on grass pea with the goal of developing varieties that fulfil its potential as a high protein, low input, resilient, climate-smart crop, suitable for small-holder farmers.

## Results

### Genome assembly and annotation

Using *Pisum sativum* L. (pea) as a standard with a genome size of 4.45 Gbp[28], we undertook flow cytometry (Supplementary Fig. 1) and estimated the genome size of grass pea genotype LS007 as 6.517 Gbp ± 0.023 Gbp (SE, $n = 3$). We used this estimate to calculate the genome sequencing depth of all our sequencing datasets.

To assemble the grass pea genome, we sequenced the European accession LS007 using the Oxford Nanopore Technology (ONT). The libraries were sequenced on the PromethION platform with subsequent loads on the same flow cell separated by nuclease flushes. In total, 296.15 Gbp of sequence passed the quality filter, representing 45.44 X coverage of a 6.5 Gbp genome of all lengths. Sequence yields per flow cell load and distributions of read lengths (post-filter) are shown in Supplementary Table 1 and Supplementary Fig. 2 respectively. After filtering for reads >5 kbp, Redbean[29] produced an assembly of 6.2 Gbp, based on 34.6X coverage. The resulting assembly contained 162,985 contigs, with a contig N50 of 155,574 bp and a GC fraction of 38.8%. Following polishing with minimap2[30] and bwa[31], incorporating 49.7X coverage of paired-end Illumina HiSeq data, this resulted in an assembly of 6.237 Gbp in 162,994 contigs with an N50 of 157,998 bp (Fig. 1a, Supplementary Fig. 3, Supplementary Table 2). The assembly and all additional annotation and sequence files are available under doi: 10.5281/zenodo.7390878[32].

Hi-C scaffolding resolved 42.7% of the total assembly into 7 chromosome scale scaffolds, and 2 sub-chromosome-scale scaffolds (contact map shown in Supplementary Fig. 4). In order to not lose sequences of potentially important genes, annotation and downstream analysis were conducted with the complete, unscaffolded assembly. The 9 largest Hi-C scaffolds are available as a separate fasta file in zenodo (doi: 10.5281/zenodo.7390878)[32].

An assessment of the polished assembly accuracy and gene space completeness was conducted through benchmarking analysis of conserved genes using BUSCO[33] against Fabales, Eudicots, Viridiplantae and Eukaryota lineages, showing completeness scores of 82.6%, 85.7%, 88.5% and 89.8% respectively (Fig. 1b).

Illumina reads were mapped to the assembly using BlobTools[34] with results shown in Supplementary Table 3. A filtered list of high-quality contigs with length >50 kb, Illumina coverage 20-100x and classification (Streptophyta only) is available as a separate file. Blob-Tools was also used to identify plastidial and mitochondrial genomes in the assembly which are also available in zenodo (doi: 10.5281/zenodo.7390878)[32].

Gene models were predicted (across the entire 6.2Gbp assembly) by an evidence-guided annotation approach incorporating RNA-Seq and cross species protein alignments (see Supplementary Method 1 and 2, Supplementary Table 4). Using RNAseq data of the three grass pea genotypes LS007, LSWT11 and Mahateora[35,36], comprising 2.5 bn paired-end Illumina reads (Supplementary Table 5), we assembled transcriptomes (Supplementary Table 6). These were unified into a non-redundant set of transcripts (Supplementary Table 7, Supplementary Table 8). Gene models were assigned transcript scores based on the coverage of matching cDNAs in the transcript assemblies. In addition, gene models were assigned protein ranks based on coverage compared to a database consisting of gene models of nine plant species (*Cicer arietinum, Cucumis sativus, Fragaria vesca, Glycine max, Malus domestica, Medicago truncatula, Prunus persica, Phaseolus vulgaris*, and *Trifolium pratense;* Supplementary Tables 9 and 10, Supplementary Method 3). Transcript and protein ranks of the gene models together were used to classify them as 'high' or 'low' confidence. These were filtered further to remove repeat associated genes and genes with near-zero transcript counts. In all, we identified 30,167 high-confidence protein-coding genes and 15,307 low-confidence protein-coding genes, which we functionally annotated using AHRD v.3.3.3.

The number of genes within each category is shown in Table 1. tRNA gene prediction was carried out using tRNAscan-SE[37], and resulted in a total of 2801 tRNA genes predicted for the polished assembly.

### Repeat structure

To obtain an unbiased estimate of the proportion of repeats in the analysed genome, we screened a subset (0.1 x coverage) of Illumina paired-end reads for repeated elements using the RepeatExplorer2 algorithm[38]. Supplementary Table 11 shows the proportion of reads classified as repeats, along with literature values reported for *L. sativus*[39]. Because this analysis was based on the raw read data, it was independent of the assembly strategy used. Ty3/Gypsy Ogre elements dominate the grass pea genome, accounting for 37.3% of the genome and representing the majority of the population of LTR-retrotransposons, which together account for 57% of the genome. Satellite repeats, estimated at 8% of the genome, are second in terms of genome abundance. They comprise several sequence families that have previously been shown to form heterochromatic chromosome bands or cluster in the centromeric regions of several *L. sativus* chromosomes[40].

Comparison of repeat proportions in the unassembled Illumina reads with those in the assembled contigs (Supplementary Method 1) revealed that many high-copy repeats were underrepresented in the assembly (Supplementary Fig. 5). This was most pronounced for all families of satellite repeats, while dispersed repeats such as Ogre elements were less affected. These results reflect the difficulty of most assemblers to bridge repeat-rich regions, which is particularly severe for long arrays of satellite repeats. Despite the difficulty in assembling the most repetitive portion of genomic DNA, annotation of the repeats in the assembled contigs showed that they were still rich in repetitive DNA. The annotated LTR-retrotransposon sequences accounted for 78.3% of the contig lengths, and the only repeat type that was underrepresented in the assembly were satellite repeats and rRNA gene clusters (Supplementary Table 11).

While the highly abundant Ogre, Athila and Maximus/SIRE elements display unimodal distributions in branching times, several less abundant classes of transposable elements (Angela, Bianca, Ikeros, Ivana, TAR and CRM) display multimodal distributions of branching times, suggesting periods of rapid divergence among specific classes of transposable elements (Fig. 2, Supplementary Fig. 6).

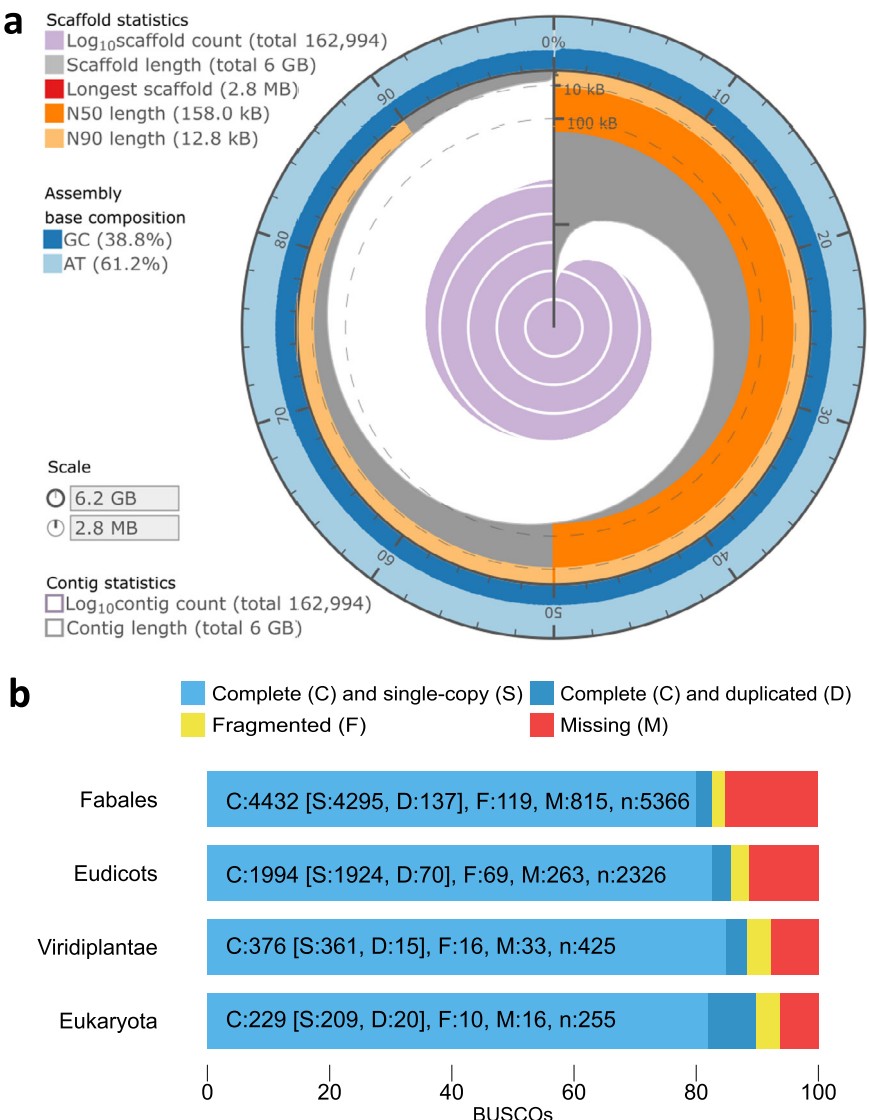

**Fig. 1 | Assembly benchmarking. a** Assembly statistics for the LS007 long-read assembly, visualised using assembly stats[95]. **b** BUSCO (v.4.0.4) assessments of the *L. sativus* LS007 genome assembly against the Fabales, Eudicots, Viridiplantae and Eukaryota datasets. Source data are provided as a Source Data file.

**Table 1 | Annotation biotype and gene confidence assignment for *L. sativus* LS007 genome assembly**

| Confidence level | Biotype | Gene count |
|---|---|---|
| High | protein_coding | 30,167 |
| High | protein_coding_repeat_associated | 6131 |
| Low | protein_coding | 15,307 |
| Low | protein_coding_repeat_associated | 6580 |
| Low | predicted gene, unknown coding | 5737 |
| Total | | 63,922 |

### Genes encoding previously characterised enzymes involved in β-L-ODAP biosynthesis

The pathway for β-L-ODAP biosynthesis in grass pea was first proposed by Malathi et al.[27] on the basis of activity assays and partial purification of the enzymes involved (Fig. 3a). The first dedicated step in β-L-ODAP synthesis is the synthesis of β-(isoxazolin-5-on-2-yl)-alanine in mitochondria. The formation of this compound from isoxazolin-5-one and cysteine is catalysed by β-cyanoalanine synthase (CAS) under high-sulphur conditions, while under low-sulphur conditions it is primarily formed from isoxazolin-5-one and O-acetyl-serine, catalysed by cysteine synthase (CS)[26,41,42]. One gene encoding CAS and at least 4 genes encoding CS have been identified from grass pea and their activities have been characterised[42]. Our assembly contains *LsCAS* located on ctg2942 (annotated LATSA3860_EIv1.0_0290570), and the *LsCS* gene described by Chakraborty et al.[43] on ctg2511 (LAT-SA3860_EIv1.0_0248570), with a highly similar copy on ctg707 (LAT-SA3860_EIv1.0_0542380). In addition to these three, our assembly contains 19 other genes that have been automatically annotated as cysteine synthases (Supplementary Data 2).

### L-DAP is present in grass pea and pea (*P. sativum*)

L-DAP has been inferred to be the immediate precursor for β-L-ODAP synthesis[24,27,44] (Fig. 3a), but its existence in plant tissues has not been confirmed directly. Using an LCMS method[36], we measured the wt/dry wt concentration of L-DAP in the shoot tips of seedlings of grass pea LS007 as 0.015% ± 0.002% w/w and Mahateora as 0.005% ± 0.0003% w/w, *P. sativum* cv. Cameor as 0.022% ± 0.001% w/w and *M. truncatula* cv. A17 as not detectable (at a detection limit of 0.005% w/w). The complex derivatisation pattern of L-DAP in a crude extract made it impossible to quantify this compound more accurately using this

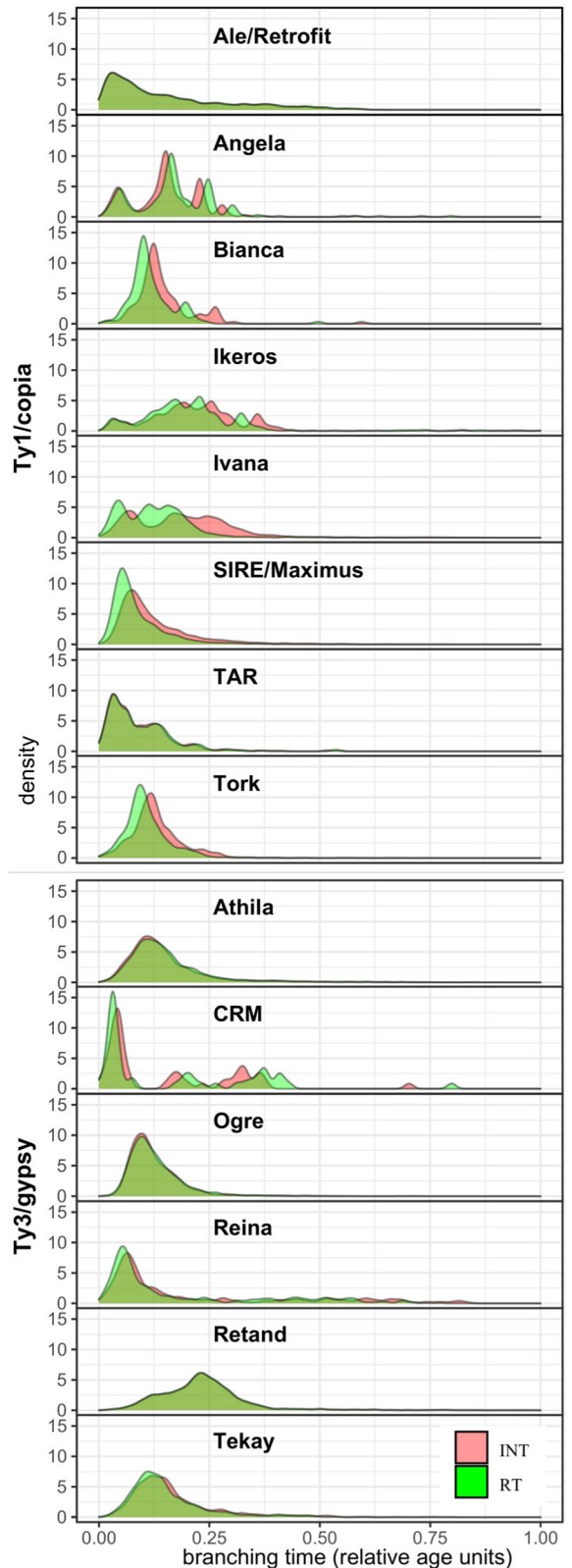

**Fig. 2 | Ages profile of Class I TE lineages.** Relative ages are inferred from the branching distribution of neighbor-joining trees (Supplementary Figure 6), built from the INT (red) and RT (light green) protein domains, respectively. Source data are provided as a Source Data file.

Formation of L-DAP using grass pea enzyme extracts in vitro has been described[25], however, this reaction occurred only under non-physiological conditions (pH 9-10) and no associated enzyme has been identified in grass pea to date. A gene, SbnB, predicted to encode an ornithine cyclodeaminase, is responsible for production of L-DAP in *Staphylococcus aureus*, a precursor of antibiotics in this organism[45]. Its closest relative in grass pea (a gene of unknown function encoding a predicted protein with 21% amino acid identity to SbnB, LAT-SA3860_EIv1.0_0016210, ctg105268) is expressed at very low levels throughout the plant (Fig. 3c).

### LsAAE3 links oxalate catabolism to β-L-ODAP biosynthesis

An oxalyl-CoA synthetase (acyl-activating enzyme 3, AAE3) has been described in *Arabidopsis thaliana*[46] and *M. truncatula*[47] as well as in *Saccharomyces cerevisiae*[48], where it is involved in the catabolism of oxalate. In *Arabidopsis* and *Medicago*, this pathway also confers partial resistance to fungal pathogens that produce oxalate crystals to disrupt plant cells *(Sclerotinia* spp.)[46,47]. The grass pea genome contains a gene encoding an enzyme with 75% amino acid identity to AtAAE3 and 88% amino acid identity to MtAAE3. This gene has been described recently in grass pea[49,50] and is found on ctg4766 of our assembly, annotated as LATSA3860_EIv1.0_0424890.

### β-L-ODAP production *in planta* involves the activity of LsAAE3 in concert with LsBOS

The final reaction in the β-L-ODAP biosynthesis pathway has been proposed to transfer the oxalyl moiety from oxalyl-CoA to the terminal amine group of L-DAP[27]. Reactions of this type (an N-acylation using a CoA-conjugated acyl donor) are commonly catalysed by enzymes of the BAHD-acyltransferase superfamily in plants. We identified all likely BAHD-AT candidates (70 genes) from grass pea by automated annotation of the genome using RNA-seq data generated from the genotypes LSWT11, LS007 and Mahateora. A phylogeny of all annotated BAHD-ATs in the LS007 genome assembly, along with BAHD-ATs in *P. sativum* and *A. thaliana* is shown in Supplementary Data 1. One clade of BAHD-ATs stood out as having expanded in legumes (Fig. 3b). Nine transcripts (labelled BAHD1-9) corresponding to this clade were present in multi-tissue transcriptome encoding BAHD-ATs from this clade, (BAHD 5, 8, 9 were so similar that they likely represent transcript isoforms of the same gene). Based on mRNA abundance in grass pea tissues (Fig. 3c), we selected four isoforms (BAHD 2, 3, 8 and 9) for cDNA amplification and heterologous expression in *Nicotiana benthamiana* using the pEAQ-HT expression system. Expression of just one of these (BAHD3) resulted in the formation of β-L-ODAP in *N. benthamiana* when L-DAP was co-infiltrated into *Agrobacterium*-inoculated leaves (Fig. 3d). This gene, which we named *β-L-ODAP synthase* (*LsBOS*), is present on contig ctg14433 in our assembly, as a 1320bp-long intron-less gene. The clade containing *LsBOS* and the other BAHD-AT genes tested is shown in Fig. 3b. None of the other candidate BAHD-ATs showed activity in forming β-L-ODAP.

We cloned the coding region of *LsBOS* from grass pea and expressed it in *Escherichia coli*. We produced and purified the recombinant enzyme and performed in vitro reactions with added substrates to test whether LsBOS acted as a BAHD acyl transferase with oxalyl-CoA and DAP as originally proposed by Malathi et al.[27] (Fig. 3e). Oxalyl-CoA was chemically synthesised using the method described by Quayle et al.[51]. However, incubation of purified LsBOS with oxalyl-CoA and L-DAP did not produce any β-L-ODAP. Moreover, LsBOS showed no activity in the reverse direction when incubated with β-L-ODAP and CoA, despite reversibility being a common feature of BAHD acyl-transferases. We investigated the activity of LsBOS further by examining the activity of LsAAE3 alone and then combined LsAAE3 with LsBOS in a coupled assay, followed by measuring β-L-ODAP production in the reactions using LCMS[36]. LsAAE3 produces oxalyl-CoA when incubated with oxalate, $Mg^{2+}$, ATP and CoA. However, when L-DAP is

method. Our results confirm the presence of small quantities of L-DAP in grass pea and pea tissues but not in the more distantly related legume species, *M. truncatula*.

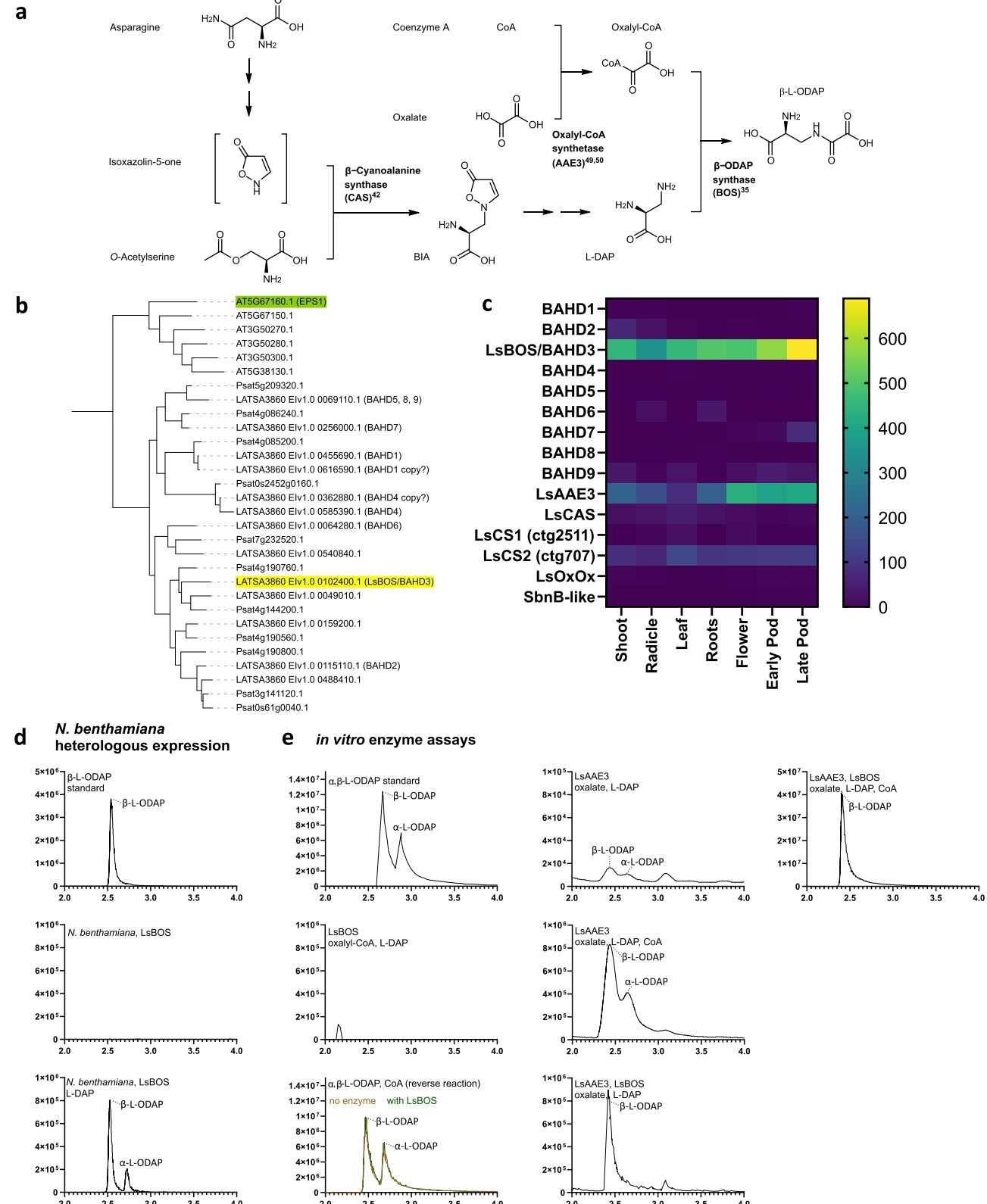

**Fig. 3 | Identification and in vitro confirmation of LsBOS. a** Grass pea β-L-ODAP biosynthesis as reviewed by Yan et al.[44] **b** Clade including EPS1 and LsBOS (highlighted) from the maximum likelihood phylogeny of BAHD-ATs of *A. thaliana, P. sativum* and *L. sativus* (full phylogeny shown in Supplementary Data 1). **c** Heatmap transcript abundance of genes of interest in *L. sativus* (in TPM). **d** LCMS spectra showing β-L-ODAP formation in *N. benthamiana* expressing LsBOS, in presence of L-DAP. **e** LCMS spectra showing α- and β-ODAP formation in in vitro assays using LsBOS, LsAAE3 and different combinations of substrates. Source data are provided as a Source Data file.

supplied instead of CoA (and at pH 8.0) small amounts of both β- and α-L-ODAP were produced in vitro. When CoA was added to this reaction, the production of β-L-ODAP and α-L-ODAP increased substantially. When LsAAE3 and LsBOS were incubated with oxalate, Mg$^{2+}$, ATP, L-DAP, β-L-ODAP production increased substantially compared to LsAAE3, oxalate, Mg$^{2+}$, ATP and L-DAP alone, but very little/no α-L-ODAP was detected. When CoA was added to the reaction involving both enzymes with oxalate, Mg$^{2+}$, ATP and L-DAP even more β-L-ODAP but no α-L-ODAP was detected.

To address the likely physiological activity of LsAAE3 in synthesising either oxalyl-CoA or β-L-ODAP in grass pea, we determined the pH optima of its two activities alongside the pH optimum for the coupled LsAAE3-LsBOS reaction in vitro. Remarkably, the pH optimum for the synthesis of oxalyl-CoA by LsAAE3 was 6.0 (Fig. 4a) with virtually no activity remaining at pH 7.0, which is close to the normal pH of the cytoplasm in plants (7.5) where LsAAE3 is thought to be located[46]. In contrast, the synthesis of β-L-ODAP and α-L-ODAP by LsAAE3 had a pH optimum between 8.0 and 9.0, similar to the optima measured for AAE3 enzymes from other plants for the synthesis of oxalyl-CoA. Activity was low at pH 7.0 but rose rapidly with pH increasing to 8.0. The pH optimum for the LsAAE3-LsBOS coupled reaction producing β-L-ODAP was 9.0 but retained substantial activity at pH 7.0, in vitro.

To confirm whether the low pH optimum of LsAAE3 for the synthesis of oxalyl-CoA was unique to the enzyme from grass pea, the pH optimum of MtAAE3 from *M. truncatula* was assayed in vitro, following expression in *E. coli* and purification. Direct measurement of oxalyl-CoA production confirmed the pH optimum of 7.0 reported by Foster et al[47] using an indirect activity assay measuring hydrolysis of ATP (Fig. 4a). Therefore, the low pH optimum of grass pea LsAAE3 for the synthesis of oxalyl-CoA would appear to be unusual and might imply that the enzyme does not form oxalyl-CoA *in planta*.

Our analyses suggested that LsBOS enhances the production of β-L-ODAP by LsAAE3 in the presence of oxalate, ATP and L-DAP. The stimulatory effect of CoA on the LsAAE3-catalysed production of β-L-ODAP and α-L-ODAP and in the LsAAE3-LsBOS coupled reaction, producing only β-L-ODAP, could be the result of CoA acting as an activator of the reaction rather than as a substrate. Hence, we explored the possible mechanism of the reaction further by determining apparent $K_m$ values for LsAAE3 using oxalate and CoA as substrates in its CoA-ligase activity at pH 6.0. The apparent $K_m$ values of LsAAE3 for oxalate in the synthesis of oxalyl-CoA was measured as 2 mM and for CoA was measured as 5 mM. In contrast, in LsAAE3's reaction with oxalate, ATP and L-DAP synthesising β-L-ODAP and α-L-ODAP at pH 8.0, the apparent $K_m$ for oxalate was measured as 600 μM, and for L-DAP was 2.5 mM, but for CoA was 1.8 nM although the low levels of ODAP synthesised limited the accuracy of the measurements possible. In the coupled reaction, synthesis of β-L-ODAP is catalysed by LsAAE3 and LsBOS with oxalate, Mg$^{2+}$, ATP, L-DAP, CoA at pH 8.0, the apparent $K_m$ was 600 μM for L-DAP, 250 μM for oxalate, and 70 nM for CoA, five orders of magnitude lower than when CoA served as a substrate in the synthesis of oxalyl-CoA at pH 6.0 (Table 2). This supports our hypothesis that CoA is acting as an activator of LsAAE3 in the synthesis of β-L-ODAP.

To confirm this, we assayed LsAAE3 activity with oxalate, Mg$^{2+}$, ATP, L-DAP with and without LsBOS in the presence of CoA and its analogues, desulpho-CoA and S-ethyl-CoA, that mimic the structure of CoA but cannot be used as substrates. Without LsBOS, CoA stimulated the synthesis of oxalyl-CoA by LsAAE3 whereas, in the presence of the analogues, no increased synthesis was observed. When LsBOS was included, substantially enhanced synthesis of β-L-ODAP was detected using CoA and its two analogues (Table 3, Supplementary Fig. 7).

We propose a mechanism for the production of β-L-ODAP in the LsAAE3-LsBOS coupled reaction where LsAAE3 acts as an adenylase in the presence of oxalate, Mg$^{2+}$, ATP and L-DAP at the physiological pH of

the cytoplasm (7.5). Oxalyl-AMP is a high-energy intermediate in acylation reactions and can further enable the transfer of the acyl group onto an amine group (as present in L-DAP) of the acyl acceptor molecule to produce the corresponding amide. This is the characteristic activity of GH3 enzymes of the AAE super family. This reaction appears to occur when LsAAE3 is incubated with oxalate, Mg$^{2+}$, ATP and L-DAP at pH 8.0 but is not regio-specific in vitro, i.e., β-L-ODAP and α-L-ODAP are produced. This reaction is accelerated by nanomolar concentrations of CoA. The presence of LsBOS provides regio-specificity to this reaction that produces β-L-ODAP. Since the acyl adenylate intermediate is held in the active site of AAE3 enzymes like LsAAE3 until the amide conjugate is formed and AMP is released, it is likely that LsBOS physically interacts with LsAAE3 in a way that enables regio-selective production of β-L-ODAP.

Using surface plasmon resonance (SPR), we investigated the LsAAE3-LsBOS interaction and observed a close association of the two enzymes. The result of this experiment suggests that LsAAE3 interacts with LsBOS with a stoichiometry of 1:1 and with association and dissociation rates of $k_{on}$ 3.67 × 10$^3$ s$^{-1}$M$^{-1}$ and $k_{off}$ 3.61 × 10$^{-4}$ s$^{-1}$, respectively (Fig. 4b). To confirm this interaction, we used co-immunoselection assays with differentially tagged LsAAE3 and LsBOS. LsAAE3 was able to co-select LsBOS and LsBOS was able to co-select LsAAE3 as shown in Supplementary Fig. 8. Interestingly inclusion of L-DAP potentiated the interaction between the two proteins. Transient expression of LsBOS (in the presence of L-DAP) resulted in the formation of β-L-ODAP in *N. benthamiana* (Fig. 3d), suggesting that the endogenous enzyme, NbAAE3, can also interact with L-DAP and LsBOS to form β-L-ODAP.

Since β-L-ODAP synthesis appears to provide a means of removing oxalate in grass pea with limited oxalyl CoA ligase activity, we challenged leaves of high β-L-ODAP (LS007) and low β-L-ODAP (Mahateora) varieties with oxalate at a range of concentrations (10 mM, 20 mM and 40 mM), excised leaves of low-β-L-ODAP genotype Mahateora developed larger lesions than leaves of high-β-L-ODAP-genotype LS007[35] (Supplementary Fig. 9).

Overall, we conclude that the synthesis of β-L-ODAP in grass pea involves a non-canonical activity of LsBOS, which interacts with LsAAE3 and modulates its activity such that it acts as an oxalyl adenylase that further transfers the oxalyl group from oxalyl-AMP to the terminal amine of L-DAP to form β-L-ODAP (Fig. 4c). The detailed molecular mechanism by which LsBOS enables regio-selective amide formation is a subject of future study. Interestingly, the apparent $K_m$ for L-DAP was substantially lower in the coupled reaction than for LsAAE3 alone, implying that LsBOS also increases the affinity for this substrate in the synthesis of β-L-ODAP. This reaction is stimulated by low levels of CoA in comparison to the alternative activity of LsAAE3 which requires relatively high concentrations of CoA and relatively low pH to form oxalyl-CoA. This latter activity has been well-characterised in other plant species and is thought to function in removing excess oxalate from plant cells, which may accumulate during the breakdown of ascorbate. However, the relative inactivity of LsAAE3 in forming oxalyl-CoA at pH 7.0 in vitro suggests that this activity is unlikely to occur under physiological conditions in grass pea.

## Discussion

The genomes of many species in the Fabeae (Vicieae) tribe, such as *Vicia faba* (13.1 Gbp), *Lens culinaris* (4.2 Gbp) and *P. sativum* (4.3 Gbp)[52] are difficult to assemble due to their size and repetitive nature. Kreplak et al.[53] recommended the use of hybrid assembly approaches for such genomes. Recent advances in long-read sequencing have increased yield and base accuracy of these technologies, making them viable for whole-genome-shotgun approaches for the assembly of multi-gigabase genomes. The greatly reduced number of individual reads needed to achieve the same coverage, compared to established short-read technologies, eases the computational complexity of long-read-only assemblies,

**a**

**oxalyl-CoA formation**

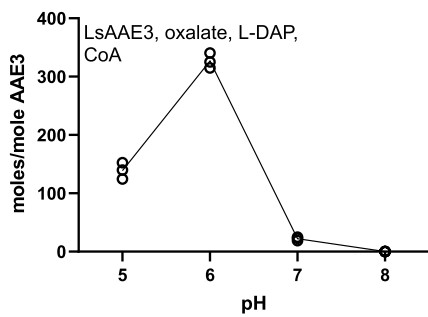

**β-L-ODAP formation**

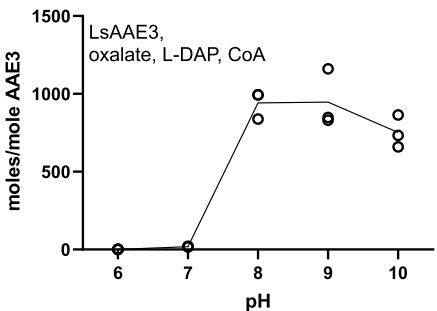

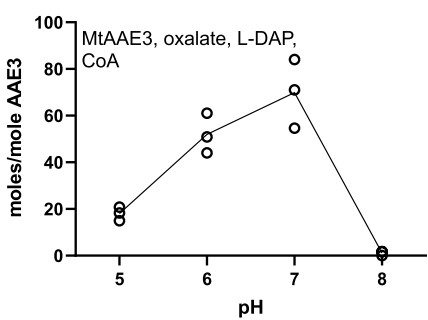

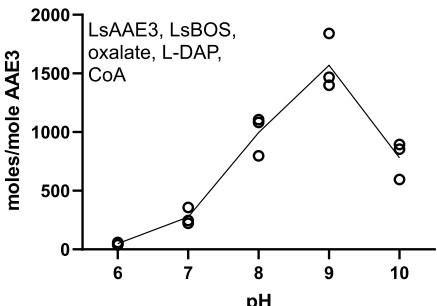

**b**

**LsAAE3 + LsBOS binding response**

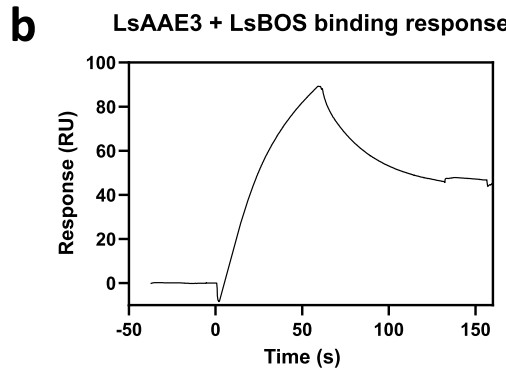

**LsAAE3 + LsBOS binding kinetics**

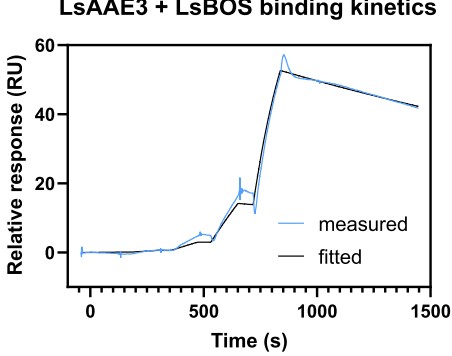

**c**

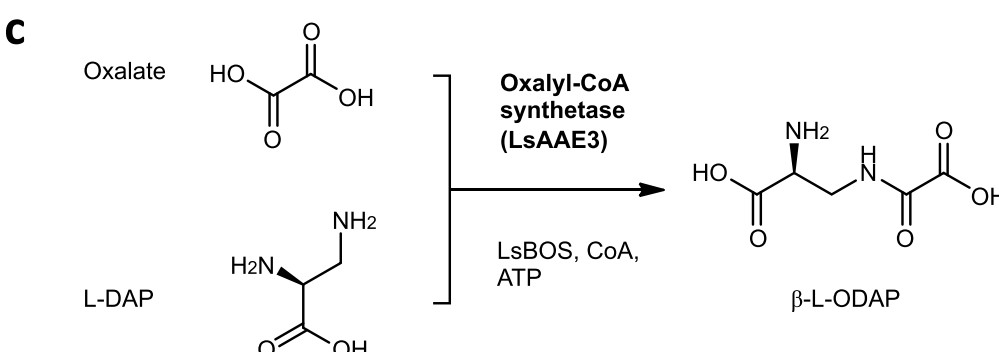

**Fig. 4 | Interaction between LsAAE3 and LsBOS shifts the pH optima of LsAAE3 activities. a** Enzyme activity in oxalyl-CoA and β-L-ODAP formation across pH. **b** SPR traces showing 1:1 binding between LsAAE3 and LsBOS. **c** Proposed final step of β-L-ODAP by LsAAE3 in complex with LsBOS and CoA. Source data are provided as a Source Data file.

allowing assemblers like redbean/wtdbg2[29], shasta[54], flye[55] and NECAT[56] to complete multi-gigabase assemblies in <10,000 CPU hours. In comparison, current hybrid assemblers that also utilise short-read data, like SPAdes[57] and MaSuRCA[58] at the initial assembly step tend to require at least tenfold higher computation times. For this reason, we opted for using a long-read-only assembler (redbean) first to assemble the grass pea genome from nanopore reads, followed by polishing with Illumina data.

**Table 2 | Kinetic parameters of LsAAE3 in the presence or absence of LsBOS**

| Reaction | Oxalate | | L-DAP | | CoA | | $k_{cat}$ (s⁻¹) |
|---|---|---|---|---|---|---|---|
| | $K_m$ (mM) | $k_{cat}/K_m$ (s⁻¹M⁻¹) | $K_m$ (mM) | $k_{cat}/K_m$ (s⁻¹M⁻¹) | $K_m$ (mM) | $k_{cat}/K_m$ (s⁻¹M⁻¹) | |
| LsAAE3 (pH 6.0) Oxalate → oxalyl CoA | 2.0 ± 0.2 | 7.5 × 10⁻² | n/a | n/a | 5.0 ± 1.0 | 3.0 × 10⁻² | (1.50 ± 0.09) × 10⁻⁴ |
| LsAAE3 (pH 8.0) L-DAP → β-L-ODAP | 0.6 ± 0.4 | 1.9 × 10³ | 2.5 ± 0.5 | 456 | (1.8 ± 0.8) × 10⁻⁶ | 6.3 × 10⁸ | 1.14 ± 0.10 |
| LsAAE3/LsBOS (pH 8.0) L-DAP → β-L-ODAP | 0.25 ± 0.05 | 1.2 × 10³ | 0.6 ± 0.1 | 1.97 × 10⁴ | (7.0 ± 1.0) × 10⁻⁵ | 1.69 × 10⁸ | 11.8 ± 0.76 |

Presented as mean ($n$ = 3) ± standard deviation.

The grass pea genome follows the recent publication of the reference genome of *P. sativum*[53], a close relative of grass pea. Assembling the genome of grass pea is rendered more difficult due to its larger size (we estimate 6.52 Gbp for cultivar LS007) and the relative lack of genetic resources such as high-density genetic and/or molecular maps that could be used for scaffolding to the pseudochromosome level.

Most species in the Fabaceae have genome sizes ranging from 0.3 Gbp to 1.5 Gbp[52]. However, diploid genome sizes within the Vicieae tribe (Fig. 5) range from 1.7 Gbp (*V. lunata*) to 13 Gbp (*V. faba*). This tribe branched from the rest of the family Fabaceae 16–25 Mya[59–61]. Some Vicieae lineages have undergone rapid genome expansion, driven by replication of repeated sequences[39]. This variation correlates strongly with the copy numbers of repeated elements[40,52]. Considerable variation in genome size may even appear within species, as Ghasem et al.[62] described for grass pea, suggesting rapid gain or loss of sequences. Ogre elements are a type of Ty3/Gypsy LTR retrotransposon first discovered in legumes[63,64], but since found in other plant families as well[65]. Ogre elements are characterised by their large size (up to 25 kbp) and the presence of an additional ORF upstream of the *gag-pro-pol* polyprotein ORF usually present in LTR retrotransposons[65]. In a survey of the genomes of 23 species within the Vicieae tribe, Ogre elements typically make up about 40% of the entire genome (22.5 − 64.7%), and Ogre-content correlates strongly with genome size[39]. This high level of repetition underlines the difficulty of assembling the grass pea genome and the necessity for long sequence reads that can span repetitive regions.

β-L-ODAP biosynthesis is linked to the metabolism of sulphur-containing amino acids (cysteine and methionine). This is noteworthy because these amino acids are also protective against β-L-ODAP toxicity in cell culture and malnutrition for these amino acids greatly increases susceptibility to neurolathyrism in humans. The first dedicated step in β-L-ODAP synthesis is the synthesis of β-(isoxazolin-5-on-2-yl)-alanine (BIA) in mitochondria[41,42] (Fig. 3a). Under high-sulphur conditions, the formation of BIA from isoxazolin-5-one and cysteine is catalysed by CAS, while under low-sulphur conditions BIA is formed at lower rates from isoxazolin-5-one and O-acetyl-serine, catalysed by CS, both members of the BSAS family[26,41,42].

The essential roles of these enzymes in cysteine formation (CS) and cyanide detoxification (CAS) make these genes difficult targets for the breeding of low-β-L-ODAP grass pea varieties. BIA is a common metabolite in legume seedling roots, with a putative role as an antifungal agent[25,66]. BIA is secreted from pea roots and likely acts as an allelochemical. These roles raise the possibility that the defence of grass pea against pathogenic soil fungi might be compromised if the production of BIA is disrupted[66]. Disrupting CAS could also lead to the hyperaccumulation of cyanide, making the enzyme a poor target for metabolic disruption to reduce β-L-ODAP accumulation.

Our results confirmed the presence of small quantities of L-DAP in grass pea and pea tissues but not in *M. truncatula*. Interestingly, the commercial low β-L-ODAP grass pea variety Mahateora had relatively low L-DAP content suggesting an early impairment in the β-L-ODAP biosynthetic pathway.

**Table 3 | Effect of non-reactive CoA-analogues on LsAAE3 oxalyl-CoA formation and LsAAE3 + LsBOS β-L-ODAP formation**

| Treatment | Oxalyl-CoA formation (LsAAE3) | | β-L-ODAP formation (LsAAE3 + LsBOS) | |
|---|---|---|---|---|
| | µmol·s⁻¹·mol LsAAE3⁻¹ | % of product with CoA | µmol·s⁻¹·mol LsAAE3⁻¹ | % of product with CoA |
| CoA | (1.84 ± 0.11) × 10⁵ | 100 % | (5.17 ± 0.11) × 10⁶ | 100 % |
| Desulpho-CoA | 13.58 ± 2.47 | 0.007 % | (1.17 ± 0.04) × 10⁵ | 2.3 % |
| S-ethyl-CoA | 10.24 ± 3.73 | 0.005 % | (1.96 ± 0.09) × 10⁵ | 3.8% |

Oxalate, Mg²⁺, L-DAP and ATP were present in all reactions. Presented as mean ($n$ = 3) ± standard deviation.

The enzymes described in this study (LsAAE3 and LsBOS) together catalyse the final step in β-L-ODAP biosynthesis. To gain further insight into the mechanism of action of LsBOS, we referred to the BAHD phylogenetic tree (Fig. 3b). The closest homologue sharing a common ancestor with LsBOS with known functionality is EPS1 from *A. thaliana*. EPS1 has recently been shown to function in the release of salicylate from an isochorismyl-glutamate conjugate formed by the activity of PBS3, a GH3 enzyme that conjugates the carboxyl group of isochorismate with the amine of glutamate[67]. PBS3, like LsAAE3, belongs to the superfamily of acyl-activating enzymes that form an acyl-adenylate intermediate, from carboxylic acid and ATP precursors with the release of PPi[68,69].

AAE3 is involved in the catabolism of oxalate in many plant species including *M. truncatula*. However, in *L. sativus* there appears to have been an evolutionary repurposing of this existing pathway, leading to the production of β-L-ODAP, perhaps providing an alternative route for removing oxalate. We propose that the production of β-L-ODAP in the LsAAE3-LsBOS coupled reaction indicates an activity of LsAAE3 as an adenylase in the presence of oxalate, Mg²⁺, ATP and L-DAP at the physiological pH of the cytoplasm (7.5), where AAE3 is localised[47]. Oxalyl-AMP is a high energy intermediate in acyl activating enzyme reactions and can interact with amine groups (as in L-DAP) to produce the corresponding amide. This activity predominates in GH3 enzymes of the AAE superfamily (of which PBS3 is a member). This reaction appeared to occur spontaneously with LsAAE3, oxalate, Mg²⁺, ATP, L-DAP at pH 8.0, but was not regio-specific in vitro, so that both β-L-ODAP and α-L-ODAP were produced. The reaction was stimulated by nanomolar concentrations of CoA. The presence of LsBOS enhances this reaction substantially and provides regio-specificity to this reaction to produce exclusively β-L-ODAP, but the catalytic mechanism awaits structural resolution of the complex. Since acyl/carboxyl-AMP intermediates are held in the active site of AAE3 enzymes until the conjugate is formed and AMP is released, it is likely that LsBOS interacts closely with LsAAE3 during the synthesis of β-L-ODAP, possibly through a physical interaction promoted by L-DAP as part of a metabolon.

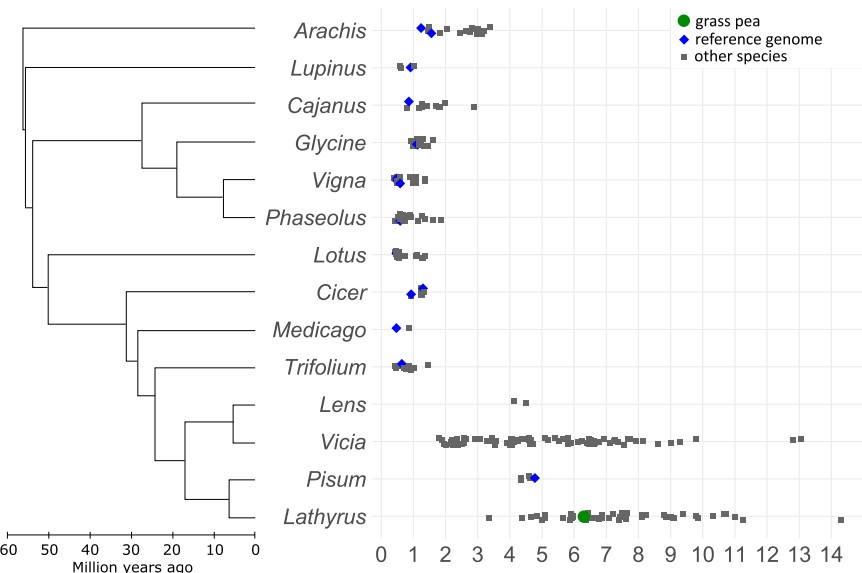

**Fig. 5 | Phylogenetic tree and haploid genome sizes of selected legume genera.** Phylogeny based on Lavin et al.[59]. Grass pea genome size based on this study, other genome sizes based on Kew c-value database[52]. Source data are provided as a Source Data file.

In this context, the enzymatic activity of LsBOS is analogous to EPS1[70]: neither enzyme serves as a canonical BAHD acyltransferase, neither enzyme produces CoA as a product, both promote acyl/carboxyl transfer via an acyl/carboxyl-AMP intermediate to an amine group, and both enhance low levels of spontaneous activity catalysed by a member of the AAE superfamily.

The different pH optima of the alternative reactions of LsAAE3 may facilitate the synthesis of β-L-ODAP by the LsAAE3-LsBOS coupled reaction. This difference in pH optima is not shown by MtAAE3, although MtAAE3 is able to synthesise β-L-ODAP in vitro when combined with LsBOS. Interestingly, the apparent $K_m$ for L-DAP was substantially lower in the coupled reaction than for LsAAE3 alone, implying that LsBOS also increases the affinity for this substrate in the synthesis of β-L-ODAP. This reaction is stimulated by low levels of CoA in comparison to the canonical activity of LsAAE3 which requires relatively high concentrations of CoA and relatively low pH to form oxalyl-CoA. However, the relative inactivity of LsAAE3 in forming oxalyl-CoA at pH 7.0 in vitro suggests that this activity is unlikely to occur under physiological conditions in grass pea. In *M. truncatula*, loss of AAE3 activity is associated with accumulation of druse crystals of calcium oxalate and increased susceptibility to oxalate-secreting phytopathogens due to the impairment of the catabolic pathway removing oxalate[47]. It is possible that the repurposing of LsAAE3 activity by LsBOS has introduced a new route for removal of oxalate through the formation of β-L-ODAP in grass pea.

The connection between these pathways raises the possibility that a disruption of β-L-ODAP-biosynthesis, particularly at the LsAAE3 stage, could lead to the hyperaccumulation of oxalate crystals[46–48]. This could negatively affect plant health and straw palatability and increase susceptibility to oxalate-accumulating necrotrophs such as *Sclerotinia* spp.

Our findings on this mechanism are at variance with the interpretations recently reported of the activity of LsAAE3 (also referred to as LsOCS[49]) and LsBOS[49,71]. These reports assume that oxalyl-CoA is the substrate for LsBOS which transfers the oxalyl group to L-DAP for the synthesises β-L-ODAP by a standard BAHD acyl transferase mechanism. However, the evidence offered for this activity of LsBOS has been based entirely on 'coupled assays' of LsBOS, in which LsAAE3 was used to synthesise oxalyl-CoA 'in situ'. All assays, including those used for LsBOS protein purification, measuring kinetic parameters and even crystallization of LsAAE3[49] were undertaken at pH 8.0, a pH at which

LsAAE3 would have been unable to synthesise oxalyl-CoA. However, all the experimental data reported by Goldsmith et al.[49,71] are consistent with the catalytic mechanism we propose, although the interpretation of the docking experiments described in these publications may require re-evaluation. The complexity of LsAAE3-LsBOS interaction also explains the results of Malathi et al.[27] who proposed an oxalyl-CoA intermediate in β-L-ODAP synthesis in grass pea but who were unable to purify a single protein with an oxalyl-CoA-dependent acyl transferase activity.

LsBOS facilitates the final reaction in the synthesis of β-L-ODAP. We were unable to detect any BOS activity for the most similar paralogs of this gene in grass pea, but it cannot be ruled out that other enzymes with BOS activity exist in grass pea. However, the high level of expression of LsBOS in grass pea tissues suggests that this is the primary enzyme involved in β-L-ODAP synthesis, making LsBOS a target for attempts to disrupt β-L-ODAP synthesis in grass pea. This may be possible to achieve through gene editing using TALEN[72] or CRISPR/Cas9[73,74] technology, or through the use of TILLING[75,76]. However, mutations conferring complete disruption of β-L-ODAP synthesis have not yet been identified, although varieties with low β-L-ODAP contents have already been developed by selection from natural germplasm[17,22,77]. The ability of LsAAE3 to synthesise β-L-ODAP and α-L-ODAP in the absence of LsBOS in vitro, may explain why no β-L-ODAP-free mutants of grass pea have yet been identified.

Mutations in genes in β-L-ODAP synthesis particularly at the LsAAE3 stage, could lead to the hyperaccumulation of oxalate crystals[46–48], potentially compromising plant health and straw palatability. Grass pea does not commonly suffer from oxalate secreting fungal pathogens such as *S. sclerotiorum* (a pathogen first reported on grass pea in 1990, under unusually damp cultivation conditions[78]). However, *Sclerotinia* spp. were more recently reported to cause extensive damage in Australia in Ceora, a low β-L-ODAP variety (0.04–0.09 % w/w)[79]. This raises the possibility that β-L-ODAP synthesis evolved in grass pea as a sink for oxalate, enabling better defence against oxalogenic fungi, following earlier mutations that shifted the pH optimum of LsAAE3, reducing its efficiency in catabolising oxalate via oxalyl-CoA. This idea is supported by relatively higher oxalate toxicity in low compared to high β-L-ODAP varieties (Supplementary Fig. 9). Effective elimination of β-L-ODAP accumulation might therefore be dependent on restoring a higher pH optimum to LsAAE3 as well as knocking out LsBOS activity.

## Methods

### Genome size estimation by flow cytometry

Grass pea genome size was estimated following the procedure described by Dolezel et al.[80]. Fresh, young leaf tissue (40 mg) of grass pea (LS007) and *P. sativum* (semi-leafless, obtained from a local market in Nairobi) was sliced finely using a scalpel blade while immersed in 2 mL of ice-cold Galbraith buffer (45 mM $MgCl_2$, 30 mM sodium citrate, 20 mM MOPS, 0.1% w/v Triton X-100, pH 7). Three biological replicates were prepared for each grass pea and pea. Supernatants were filtered through one layer of Miracloth (pore size 22–25 μm). One aliquot of 600 μL was prepared from each replicate, along with three grass pea + pea mixes at 2:1, 1:1 and 1:2 ratios, respectively. Propidium iodide was added to each tube to a concentration of 50 μM. Reactions were incubated for 1 h on ice before measuring on a FACSCantoII flow cytometer (Becton Dickinson) with flow rates adjusted to 20–50 events/s. Results were analysed using FCSalyser (v. 0.9.18 alpha), using the gating strategy shown in Supplementary Fig. 10. Grass pea genome size was estimated from the mixed sample by dividing the mean of the PE-A peak corresponding to grass pea nuclei by the mean of the PE-A peak corresponding to pea nuclei and multiplying by 4.45 Gbp, the estimated genome size of pea[53].

### Illumina sequencing

Seeds of grass pea (*L. sativus*) were obtained from Kings Seeds, Coggeshall, UK and underwent six generations of single-seed descent at the John Innes Centre, Germplasm Research Unit (GRU). This accession, named LS007 is of European origin, white-flowered, with fully cream-coloured, large and flattened seeds. Genomic DNA was isolated from the etiolated seedlings using a modified CTAB protocol and subsequently, high molecular weight DNA purified using the Qiagen MagAttract kit. Paired end (PE) sequencing was carried out using PCR-free libraries on five lanes on the Illumina HiSeq 2500 platform following the manufacturer's directions.

RNA was extracted from 7 tissues of grass pea genotype LSWT11: shoot tips and root tips of 10-day-old seedlings, whole leaflets and roots of 3-week-old plants and flowers of 3-month-old plants using the RNeasy® Plant Mini kit (Qiagen, Hilden, Germany) and early pods and late pods of 3-month-old plants using the Spectrum Plant Total RNA kit (Sigma-Aldrich, St Louis, Missouri, USA). RNA samples were reverse transcribed into cDNA and sequenced using Illumina HiSeq 2500 single end sequencing, followed by transcriptome assembly using Trinity v2.1.1[81]. Read counting was conducted using RSEM v1.2.20[82].

### Oxford Nanopore sequencing

Two methods of DNA extraction and three methods of DNA processing were used to optimize yield and read length profile of the PromethION sequencing. For Circulomics Nanobind Plant Nuclei Kit extraction (CN), fresh, aseptic LS007 shoot tissue (0.5 g) was ground in liquid nitrogen using a mortar and pestle. The resulting powder was resuspended by vortexing in 10 mL of nuclei extraction buffer (0.35 M Sorbitol, 100 mM Tris pH 8.0, 5 mM EDTA and 1% β-mercaptoethanol) and then filtered through Miracloth. The filtrate was centrifuged at 750 × *g* for 15 min at 4 °C and the pellet was resuspended in nuclei extraction buffer (with the addition of 0.4% Triton). The wash was repeated before centrifugation and resuspension in 1 mL of nuclei resuspension buffer (0.35 M Sorbitol, 100 mM Tris, 5 mM EDTA). The suspension was then centrifuged at 750 × *g* for 15 min at 4 °C and the supernatant was removed. The resulting nuclear pellet was taken into the Circulomics Nanobind Plant nuclei Kit according to the manufacturers protocol. The resulting DNA had a peak fragment size >60 kbp.

For Qiagen DNeasy Plant Mini Kit extraction (QD), 0.5 g of fresh, aseptic LS007 shoot tissue was ground under liquid nitrogen using a mortar and pestle and resuspended in 2 mL of buffer AP1 (Qiagen) with 20 μL of RNase I (Qiagen). This was incubated at 65 °C for 10 minutes, before aliquoting into 5 tubes and proceeding with the Qiagen DNeasy Plant Mini Kit protocol. Eluted DNA was pooled into a single tube and had a peak fragment size of 48 kbp. The process of sample preparation was iteratively optimized to achieve both high total sequence yield and long read length.

To increase the concentration of DNA for input into the Short Read Eliminator and to reduce the loss of DNA in library preparation, all except the DNA Qiagen DNeasy-extracted samples were pre-incubated with 0.8x Ampure XP (Beckman Coulter). Beads were washed twice in 80% ethanol and DNA was eluted from beads with TE buffer (10 mM Tris HCl pH 8.0, 1 mM EDTA).

Starting with the second loading of flowcell FC1, all except the Qiagen DNeasy-extracted samples were needle-sheared 30 times with a 26-gauge needle to reduce the amount of very high molecular weight DNA, which can cause blocking and therefore an artificially low N50.

Starting with the 3rd loading of flowcell FC1, all samples were subjected to a Circulomics Short Read Eliminator (Circulomics) treatment to reduce the number of short fragments.

All libraries were prepared using the Genomic DNA by Ligation kit (SQK-LSK109) (Oxford Nanopore Technologies) following the manufacturer's procedure. Libraries were loaded at between 250–400 ng onto R9.4.1 PromethION Flow Cells (Oxford Nanopore Technologies) and run on a PromethION Beta sequencer. Due to the rapid accumulation of blocked flow cell pores or due to apparent read length anomalies on some runs, flow cells used in runs were treated with a nuclease flush to digest blocking DNA fragments before reloading with fresh library according to the Oxford Nanopore Technologies Nuclease Flush protocol, version NFL_9076_v109_revD_08Oct2018.

### Data processing and assembly

Fast5 sequences produced by PromethION sequencing were base-called using the Guppy high accuracy basecalling model (dna_r9.4.1_450bps_hac.cfg) and the resulting fastq files were quality filtered by the basecaller. Passed Fastq files from all five sequencing runs were pooled and assembled using Redbean[29] (previously wtdbg2, version 2.2), excluding short reads <5 kb, leaving a coverage of 34.6X (assuming a 6.5 Gbp genome). This assembly was polished with minimap2 (v. 2.17)[30] using the original nanopore dataset in fasta format, followed by polishing using bwa (v. 0.7.17)[31], incorporating paired-end Illumina HiSeq data.

### Hi-C scaffolding

A sample of multiple leaves of one LS007 plant was snap-frozen in liquid $N_2$ and ground to a fine powder using mortar and pestle. Subsequently the sample was homogenised, cross-linked and shipped to Phase Genomics (Seattle, USA) who prepared and sequenced an in vivo Hi-C library. The nanopore assembly was scaffolded using Hi-C data using Juicer[83] (version 1.6) followed by 3D-DNA[84] (release 201008-cb63403). The scaffolds were loaded into Juicebox[85] (version 2.13.07) to manually resolve visible issues, bringing together scaffolds into their respective chromosomes.

### Assembly annotation

The gene annotation pipeline[86] used to annotate the LS007 genome assembly is detailed in Supplementary Methods 1, 2 and 3.

### BlobTools analysis

The Illumina HiSeq reads that had been previously used for polishing the assembly were mapped to the assembly using bwa mem (bwa v0.7.17)[31]. The sam file was processed in samtools 1.9[87]. The NCBI nucleotide database was downloaded on (21/Oct/2022) and the assembly was aligned against this using (blast+ v2.9.0)[88]. Blobtools 1.1.1[34] was used to assess the accuracy of the assembly. The poor quality contigs were filtered according to the criteria to include only Streptophyta, coverage between 20-100x and contig length longer than 50 kb.

## BUSCO analysis

Gene space completeness was assessed using BUSCO v.4.0.4_cv1[33] and the odb10 databases for eukaryta, viridiplantae, eudicots and fabales, employing default parameters.

## Analysis of BAHD acyl transferases and their phylogenetic relationships

To identify BAHD acyl transferase proteins we used blast+2.10 to identify homologous sequences in *L. sativus, P. sativum* and *A. thaliana*[88]. Each protein sequence was checked for the presence of two chloramphenicol acetyltransferase-like domains (IPR023213) using InterProScan[89] and aligned using MUSCLE 3.8[90]. The alignment was edited using Jalview 2.1[91]. We used Maximum Likelihood implemented in RAxML-8.2.12 for phylogeny inferences[92].

## Repeat analysis

The proportion of repeats in the genome was estimated by analysing unassembled Illumina reads to avoid assembly-related bias in the representation of high-abundance repeats. Illumina HiSeq paired-end data were preprocessed (trimmed, quality-filtered, cutadapt-filtered, and interleaved), downsampled to 0.1-fold coverage, and analysed with the RepeatExplorer2 pipeline, which uses a graph-based read-clustering algorithm to identify repeats[38]. The analysis was performed with default parameters on the ELIXIR_CZ RepeatExplorer Galaxy Server (https://repeatexplorer-elixir.cerit-sc.cz).

The representation of highly and moderately repeated sequences (> 0.01% of the genome) in the assembly was assessed by comparing their proportions in the assembly with the proportions estimated by RepeatExplorer2 analysis of unassembled reads[53]. The assembled contigs were split into 120 bp fragments and compared to sequences of repeat clusters previously generated using RepeatExplorer2 using BLASTN (program parameters -e 1e-20 -W 11 -r 2 -q -3 -G 5 -E 2). Each fragment was assigned to a single cluster based on its best hit (or to no cluster if it did not yield a hit). The proportion of Illumina reads in each cluster relative to the total number of reads analysed provided an estimate of the abundance of the corresponding repeats. Accordingly, the proportion of assembled sequence fragments assigned to each cluster relative to the total size of the assembly provided an estimate of the representation of the same repeat in the assembly.

Annotation of the repeats in the assembled contigs was performed using a combination of the following tools implemented on the RepeatExplorer Galaxy Server. Transposable element sequences encoding conserved protein domains were identified based on their similarities to the REXdb database[93] using DANTE. The full-length LTR-retrotransposon sequences were annotated using the DANTE_LTR tool, which combines the results of DANTE with similarity- and structure-based identification of LTR-retrotransposon signatures such as long terminal repeats (LTRs), primer binding sites (PBSs), and target site duplications (TSDs). The full-length LTR-retrotransposons identified were also used to create a reference database for similarity-based annotation of repeats in the assembly. This also included consensus sequences of repeats provided by RepeatExplorer2 and a custom database of Fabeae satellite DNA sequences compiled from our previous studies[39,40].

## Heterologous gene expression

β-L-ODAP synthase candidate genes were amplified from cDNA (primer sequences given in Supplementary Table 12) and assembled into pEAQ-HT expression vectors in *E. coli* strain DH5α using the Gateway™ cloning system. Following liquid culturing in LB medium, the vector was isolated and transformed into *A. tumefaciens* strain GV3101 pMP90 using electroporation. *A. tumefaciens* cultures were suspended in agroinfiltration solution (10 mM MgCl$_2$, 10 mM 2-(N-morpholino) ethanesulfonic acid (MES) and 200 μM acetosyringone in distilled water, adjusted to pH 5.6 using potassium hydroxide), at an O.D. of 0.2

and infiltrated into the abaxial side of 4-week old *N. benthamiana* plants using a syringe, filling >90% of mesophyll intercellular spaces. After three days, leaves were infiltrated with 1 mM L-DAP in water (pH 6 adjusted with KOH) or water (mock). Three leaves on two plants each were treated with each expression vector/substrate combination, with each leaf being treated as a biological replicate. Entire leaves were harvested five days after agroinfiltration, flash-frozen in liquid nitrogen, freeze-dried and ground using steel beads, followed by β-L-ODAP-quantification using LCMS[36].

## Cloning, expression and purification of LsAAE3 and LsBOS

*L. sativus* LS007 seeds were germinated on FP or MS-agar plates. mRNA was extracted from 5–7 day old shoots and cDNA synthesised using superscript IV reverse transcriptase (Invitrogen). Genes encoding AAE3, BOS and their closest homologues, were amplified from *L. sativus* cDNA using the following sets of Gateway-adapted primers:

*L. sativus* BOS: 5′-ATGAGTTCCATCCAAATCCTCTCCAC-3′ and 5′-TCAACCAGAAGCAGCATCCATAAAC-3′

*L. sativus* AAE3: 5′-ATGGAAACCGCAACCACCCTCAC-3′ and 5′-TCAAACTTTAGAAACAAAGTGTTC-3′

Amplified cDNAs were cloned into pDEST17 (N terminal His tag expression vector; Lifetech), transformed into *E. coli* ArcticExpress (DE3) competent cells (Agilent Technologies) and plated on LB with 100 μg/mL ampicillin and 10 μg/mL gentamycin. After overnight incubation at 30 °C, individual colonies were transferred to 5 mL LB/ampicillin and grown overnight at 30 °C. These cultures were used to inoculate 50 mL LB/ampicillin in 250 mL flasks (1:50 dilution) and grown at 30 °C to $OD_{600} = 0.6–0.8$. Cultures were cooled to 10 °C then protein expression induced by the addition of IPTG to 0.1 mM. Cultures were grown at 10 °C for 24 hours with shaking at 300 rpm.

Cells were pelleted (4000 rpm, 4 °C, 20 mins), resuspended in 1 mL Buffer A1 (50 mM HEPES pH 8.0, 50 mM glycine, 0.5 M NaCl, 30 mM imidazole, 5% v/v glycerol, EDTA free protease inhibitor tablet – 1 tablet/50 mL A1 buffer) and lysed using a tissue homogeniser (Avestin Emulsiflex) with a homogenizing pressure of 15,000 psi.

For small scale purification, cell debris was removed by centrifugation at 15,000 × *g* for 10 min. at 4 °C. His-tagged protein was purified using a HisPur Ni-NTA spin column (Thermo Scientific) and desalted with an Amicon Ultra-0.5 filter unit (Sigma-Aldrich) in accordance with manufacturer's instructions.

For large scale purification, (1 litre culture for LsAAE3; 6 litre culture for LsBOS), cell debris was removed by centrifugation at 45,000 × *g* for 60 min. His-tagged protein was purified on a 5 mL Ni-NTA column. Purified protein was desalted and concentrated using a Vivaspin 30 K centrifugal concentrator (Sartorius) according to manufacturer's instructions. Protein was quantified by Direct Detect infrared spectrophotometry (Merck Millipore) and purity assessed by SDS-PAGE and InstantBlue Coomassie stain (Abcam).

All assays reported used proteins which retained the His-Tags on LsBOS, MtAAE3 and LsAAE3. Every assay was repeated on a small scale to compare the activity of the proteins with and without their His-tags and no differences were observed. To create cleavable tags, a TEV cleavage site (GAAAACCTGTATTTTCAG) was introduced between the His-tag and the ATG of the mature coding region[94]. His-tagged proteins were purified and desalted as described above. N-terminal tags were cleaved at 4 °C overnight in 20 mM HEPES pH7.5, 150 mM NaCl, using His-tagged TEV protease at a ratio of 100 μg protein to 50 μg protease. Proteins were passed over a HisPur Ni-NTA spin column once more and the flow through containing the cleaved enzyme was desalted and concentrated with an Amicon Ultra-0.5 filter.

## Synthesis of oxalyl-CoA

Oxalyl-CoA was synthesised by ester interchange between thiocresyloxalic acid and Coenzyme A (both from PlantMetaChem, Gießen, Germany) at pH 7.2, followed by ether extraction to remove the by-

product thiocresol[51]. Quality control of the purified oxalyl-CoA by strong anion exchange (SAX) chromatography and MALDI-ToF confirmed 64% oxalyl-CoA with 36% residual CoA (Supplementary Fig. 11).

## Enzyme assays

Enzyme assays were carried out in a total volume of 100 μL and were incubated at room temperature for 1 hour. Purified enzymes were used at a concentration of 5–7 ng/μL. Reactions were carried out in 100 mM Tris pH 8.0 and contained 2 mM DTT, 5 mM ATP, 10 mM MgCl₂, and varying concentrations of sodium oxalate (0.06 – 0.3 mM), DAP (0.05 – 0.3 mM) and CoA (5 nM – 0.1 mM). CoA analogues, S-ethyl-CoA and desulpho-CoA (Jena Bioscience) were used at a concentration of 5 μM.

## Determination of β-L-ODAP, α-L-ODAP, L-DAP, CoA and oxalyl-CoA by LC-MS

For the quantification of L-ODAP isomers, 20 μL samples of reaction mix were derivatised using AccQ-Tag reagent (Waters, Milford, MA, USA) following the manufacturer's instructions. Derivatised samples were diluted 1:100 in 0.1% (w/v) formic acid before LCMS analysis. β-L-ODAP (Lathyrus Technologies, Hyderabad, India) and L-DAP standards were prepared and quantified using a Xevo triple quadrupole TQ-S instrument (Waters, Milford, MA, USA)[36]. An aliquot of this standard was dissolved in dH₂O and repeatedly freeze-thawed to form an equilibrium between β-L-ODAP and α-L-ODAP. This standard was used for the identification of β-L-ODAP and α-L-ODAP in samples, while the pure β-L-ODAP standard was used for quantification of β-L-ODAP.

For the quantification of CoA and oxalyl-CoA, standards and samples were quantified using a Xevo triple quadrupole TQ-S instrument. Separation was on a 100 × 2.1 mm 2.6 μm 100 Å Kinetex EVO C18 column (Phenomenex) using the following gradient of methanol (solvent B) versus 50 mM formic acid adjusted to pH 8.1 with 25% ammonium hydroxide in water, run at 0.3 mL min⁻¹ and 40 °C: 0–7 min, 0–10% B; 7–10 min, 10–100% B; 10-11 min 100% B; 11-12 min, 100-0% B; 12–15 min, 0% B. During the first three minutes of the analysis, the outflow from the chromatographic column was discharged to minimize the entry of salts from samples into the MS. Detection was by positive mode electrospray MS. CoA and oxalyl-CoA were detected using multiple reaction monitoring (MRM) involving the corresponding parent ion and its respective daughter ions. For CoA: 768 and 428, 261, 136, 132, 88; for oxalyl-CoA: 840 and 333, 136, 133, 105. Further instrument settings were as follows: curtain gas, 35 psi; collision gas flow rate, medium; ion spray voltage, 4.5 kV; temperature 400 °C; ion source gas, 60 psi; and entrance potential, 10 V. The declustering potential, the collision energy and the collision cell exit potential were optimized individually using standards and the automated method development tool (Intellistart) provided with Waters' MassLynx software. The test standards were coenzyme A (Sodium salt, Merck) and oxalyl-CoA (this study).

L-DAP was measured using the same procedure, using three biological replicates, as well as one replicate each at wt/dry wt spiking levels of 0.001%, 0.005% and 0.025%. The mass transition of the double-AccQTag-derivatised ion of 445.1 u > 171.1 u was used to measure L-DAP.

## Surface Plasmon Resonance

SPR was used to investigate the interactions between LsBOS and LsAAE3. Experiments were performed using the Biacore 8 K (Cytiva) with a series S sensor chip CM5 (Cytiva) and a running buffer of 10 mM Hepes pH 7.4, 150 mM NaCl and 0.05 % tween 20 and a temperature of 25 °C. Initial pH scouting experiments were run which indicated that pH 4 was the optimum pH to use for immobilisation. A standard amine coupling approach was used to activate the chip with reagents 1-ethyl −3-(3-dimethylaminopropyl)carbodiimide (EDC) and N-hydroxy succinimide (NHS) then LsBOS, at a concentration of 20 nM in 10 mM acetate pH 4.0, was injected over Flow Cell 2 (FC2) for 210 s at

10 μL/min (leaving FC 1 blank as the reference) before blocking flow cells with ethanolamine. LsBOS was immobilised with a response of ~1700. Interaction between LsAAE3 and LsBOS was observed by injecting 1 μM LsAAE3 over FC1 and FC2 for 60 s at a flow rate of 50 μL/min before switching to buffer only flow.

A range of regeneration solutions were tested and 10 mM acetate at pH 4 was found to be the best for removing the bound LsAAE3. However, this was still not ideal, so it was decided to use a single cycle kinetics approach where no regeneration is used between analyte injections. Three start-up cycles were run using only buffer then a blank run with five zero concentration injections of LsAAE3. LsAAE3 was then injected at five increasing concentrations of 2.4, 12, 60, 300 and 1500 nM each with a contact time 120 s and a flow rate 30 μL/min. At the end of all injections buffer was flowed for 600 s to record the dissociation. A concentration dependent response could be seen confirming the interaction. The data were processed and analysed using Biacore Insight Evaluation Software with double-referenced subtraction of the data and then fitted to a simple Langmuir binding model. The association rate ($k_{on}$) was determined to be $3.67 × 10^3$ s⁻¹M⁻¹ dissociation rate ($k_{off}$) as $3.61 × 10^{-4}$ s⁻¹ giving an dissociation equilibrium constant ($K_D$) of 98 nM.

AAE3 activity was determined in the absence of other enzymes and by a coupled enzyme assay[47]. Each reaction contained: 10 μL E. coli extract, 100 mM Tris pH 8.0, 2 mM DTT, 5 mM ATP, 10 mM MgCl₂, 0.5 mM CoA, 0.4 mM NADH, 1 mM phosphoenol-pyruvate, 300 μM sodium oxalate, 10 units each myokinase, pyruvate kinase and lactate dehydrogenase, deionised water to 100 μL. Activity was measured by reduction in OD 340 nm over time. LsBOS activity was measured in reactions containing 100 mM Tris pH 8.0, 2 mM DTT, 5 mM ATP, 10 mM MgCl2, 0.5 mM CoA, 300 μM sodium oxalate, 50 μM DAP, and 10 μL each of AAE3 and LsBOS E. coli expression extracts in various combinations, made up to 100 μL with deionised water. Amounts of β-L-ODAP and α-L-ODAP produced were measured using an LCMS procedure[36].

## Co-immunoprecipitation

LsBOS with a N-terminal His-tag was expressed, extracted and purified as described above. LsAAE3 with pICSL30023 as an N-terminal S-Tag with pPGN-C as a backbone was expressed and extracted as above (but not purified).

LsAAE3 and/or LsBOS E. coli expression extracts (10 μL each) were mixed with 100 mM Tris pH 8.0, 2 mM DTT, 5 mM ATP, 10 mM MgCl₂, 0.5 mM CoA, 300 μM sodium oxalate, with and without the addition of 50 μM L-DAP, and made up to 100 μL with deionised water, followed by 1 h incubation at room temperature.

For reciprocal immunoprecipitation, protein complexes were precipitated by adding 1 μL recombinant Anti S-Tag Rabbit Monoclonal Antibody (Abcam, ab180958, EPR 12996, lot GR148713-5, final concentration 0.28 μg/mL) and incubating the mix for 1 hour at room temperature, followed by the addition of 40 μL Protein A agarose suspension (Sigma) and incubation at room temperature for 1 hour. The complex was washed three times with 10 x volume of PBS. The washed precipitate was resuspended in 40 μL SDS sample buffer and 10 μL aliquots were analysed by SDS PAGE. Western blotting was carried out using mouse anti-His-tag antibody (Fisher Scientific) MA1-21315, lot XD343962 (final concentration 1 μg/mL) as the primary antibody and Goat Anti-Mouse IgG (FC Specific), Sigma A0168, Batch 00000962/6 (final concentration 0.4 – 1.1 μg/mL) as second antibody, in accordance with the manufacturer's instructions.

Separately, protein complexes were purified at room temperature using a HisPur Ni-NTA spin column (Thermo Fisher) in accordance with manufacturer's instructions. Eluted proteins were concentrated and desalted using an Amicon Ultra Centrifugal Filter Unit (Merck). 10 μL aliquots were analysed by SDS PAGE. Western blotting was carried out using Anti S-Tag Rabbit Monoclonal Antibody (Abcam, ab180958, EPR

12996, lot GR148713-5, final concentration 0.028 μg/mL) as the primary antibody and Goat Anti-Rabbit IgG (FC Specific) Sigma A0545, Lot 069M4835V (final concentration 0.4 – 1.1 μg/mL) as second antibody, in accordance with the manufacturer's instructions.

Gels were run alongside NEB Colour Prestained Protein Standard, Broad Range (10 – 250 kDa). Blots were visualised using a SuperSignal west Pico PLUS chemiluminescent kit (Thermo scientific) and an ImageQuant 800 camera (Amersham).

### Oxalate toxicity assay
Leaflets of 4-week old grass pea plants, grown on sterile FP-medium in controlled environment conditions, were cut off and their bases submerged in oxalic acid solutions (10 mM, 20 mM, or 40 mM adjusted to pH 4.0 with KOH) or water adjusted to pH 4.0 with HCl. Photographs were taken following incubation in the controlled environment chamber for 24 h.

### Reporting summary
Further information on research design is available in the Nature Portfolio Reporting Summary linked to this article.

## Data availability
LS007 genomic PromethION and Illumina sequencing data generated in this study have been deposited in the European Nucleotide Archive under accession PRJEB33571. Illumina RNAseq and HiC raw data generated in this study have been deposited in the NCBI Gene Expression Omnibus under accession GSE223956. The genome assembly and annotations generated in this study have been deposited on Zenodo [https://zenodo.org/record/7390878]. All plant materials used in this article are available from the corresponding author. Source data are provided with this paper.

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

## Acknowledgements

This work was supported by a John Innes Centre Institute Development Grant awarded to C.M., the Biotechnology and Biological Sciences Research Council (BBSRC) Detox Grass pea project (BB/L011719/1) awarded to C.M. and T.L.W., the BBSRC SASSA UPGRADE project (BB/R020604/1) awarded to C.M. and S.K., the BBSRC Institute Strategic Programmes 'Understanding and Exploiting Plant and Microbial Secondary Metabolism' (BB/J004596/1) and 'Molecules from Nature' (BB/P012523/1) which supported C.M., T.L.W., B.S. and the Nottingham Future Food Beacon of Excellence supporting L.Y. and M.L. and the Templeton World Charity Foundation, Inc. (TWCF0400), awarded to C.M., S.K. and B.K. P.M.F.E.'s studentship was funded by the John Innes Foundation's Student Rotation programme. None of the funding bodies were involved in the design of this study, the collection or analysis or interpretation of data, or in writing the manuscript. The Galaxy server that was used for some calculations is in part funded by Collaborative Research Centre 992 Medical Epigenetics (DFG grant SFB 992/1 2012) and German Federal Ministry of Education and Research (BMBF grants 031 A538A/A538C RBC, 031L0101B/031L0101C de.NBI-epi, 031L0106 de.STAIR (de.NBI)). Computational and data storage facilities were in part provided by the ELIXIR CZ Research Infrastructure Project (Czech Ministry of Education, Youth and Sports grant no. LM2018131). We thank the JIC Chemistry platform facility and staff for their contribution to this publication. Nanopore sequencing was performed in DeepSeq at the University of Nottingham. We acknowledge the eResearch team at Queensland University of Technology (QUT) for their support and assistance to access bioinformatics tools and computational infrastructure. We acknowledge Oxford Nanopore Technologies and Daniel Fordham for their generous advice and support regarding bioinformatics using PromethION data. We acknowledge Guru Radhakrishnan, Tjelvar Olsson, Matthew Hartley, Shabhonam Caim for their vital support and advice in bioinformatics and data handling. We thank Prof Stephen Bornemann for his insight and advice on the enzymology and chemistry of β-L-ODAP formation. We also acknowledge the support of Nitika Mukhi for help with protein purification, and Noel Ellis for his helpful advice on the manuscript.

## Author contributions

Ch.M. prepared DNA and conducted PromethION sequencing. A.S. oversaw Illumina sequencing for genome polishing. I.N. prepared and polished the genome assembly. I.N., J.C., A.S., P.P. and P.M.F.E. performed assembly benchmarking and QC. R.H.M.W., M.Vic. and B.S. performed Hi-C scaffolding. J.D.M., G.G.K., J.C., and J.H. performed RNAseq and genome annotation. P.N. and J.M. performed repeat analyses and annotation. P.M.F.E. identified gene candidates and performed expression analysis. M.Vig. prepared the phylogeny of BAHD-ATs. A.E., M.R., P.M.F.E. and Z.J. designed and performed the biochemistry experiments. C.E.M.S. and A.E. performed the SPR experiments. A.E. performed the Co-IP experiment. P.M.F.E. performed the oxalate toxicity assay. St.M. synthesised oxalyl-CoA. C.Y.K. and J.K.W. contributed to the design and interpretation of the enzyme activity characterization experiments. P.M.F.E., C.M., D.S., Sa.M., L.Y., M.L., B.K., T.L.W. and S.K. designed and supervised the work and the preparation of the manuscript. All authors contributed to the development of the manuscript. All authors read and approved the final manuscript.

## Competing interests

The authors declare no competing interests.

## Additional information

**Peer review information** : *Nature Communications* thanks Steven Cannon, Michael Wink and Quanle Xu for their contribution to the peer review of this work. Peer reviewer reports are available.

[1]John Innes Centre, Norwich Research Park, Colney Lane, Norwich NR4 7UH, UK. [2]Biosciences eastern and central Africa International Livestock Research Institute Hub, ILRI campus, Naivasha Road, P.O. 30709, Nairobi 00100, Kenya. [3]Queensland University of Technology, 2 George St, Brisbane City, QLD 4000, Australia. [4]National Institute of Agricultural Botany, 93 Laurence Weaver Road, Cambridge CB3 0LE, UK. [5]School of Traditional Chinese Medicine, Capital Medical University, You An Men, Beijing 100069, PR China. [6]Earlham Institute, Norwich Research Park, Colney Lane, Norwich NR4 7UZ, UK. [7]School of Life Sciences, University of Nottingham, University Park, Nottingham NG7 2RD, UK. [8]Institute of Plant Molecular Biology, Biology Centre CAS, Branisovska 31, Ceske Budejovice CZ-37005, Czech Republic. [9]Research and Innovation Centre, Fondazione Edmund Mach, Via Edmund Mach 1, 38098 San Michele all' Adige (TN), Italy. [10]Whitehead Institute for Biomedical Research, Cambridge, MA 02142, USA. [11]Department of Biological Engineering, Massachusetts Institute of Technology, Cambridge, MA 02139, USA. [12]Department of Biology, Massachusetts Institute of Technology, Cambridge, MA 02139, USA. [13]Global Crop Diversity Trust, Platz der Vereinten Nationen 7, 53113 Bonn, Germany. [14]International Center for Agricultural Research in the Dry Areas, Avenue Hafiane Cherkaoui, Rabat, Morocco. [15]Future Food Beacon of Excellence, University of Nottingham, NG7 2RD Nottingham, UK. [16]Norwich Institute for Sustainable Development, School of International Development, University of East Anglia, Norwich NR4 7TJ, UK. ✉e-mail: peter.emmrich@jic.ac.uk

