## [Peer Review File · Nature Communications]

Genomics and biochemistry in *Lathyrus sativus* reveal a metabolon is key to β -L-ODAP biosynthesisReviewers' Comments:

Reviewer #1:

Remarks to the Author:

This manuscript reports the genome sequence and gene annotation of *Lathyrus sativus*, and the identification of components of the biosynthetic pathway for catalysis of the neurotoxin, β -L-oxalyl-2,3-diaminopropionic acid (β -L-ODAP).

The sequencing of the genome is noteworthy in itself, due to the large genome size (approximately 6.5 Gbp). The large genome is typical of genera in the Viciae tribe, including the several that include important pulse crops: *Pisum*, *Lathyrus*, *Vicia*, and *Lens*.

The biochemical work to identify components of the β -L-ODAP pathway is also important and valuable. Although the biosynthetic pathway is complex and variable, I found the analysis presented in the manuscript to be convincing. The analysis includes expression analysis, heterologous expression of candidates in *Nicotiana benthamiana*, tests of enzymatic activity in vitro, and tests of expression in planta in *L. sativus* and several relatives that don't produce β -L-ODAP. The authors propose BAHD-acyltransferase (LsBOS) as a primary target for producing low-ODAP lines of grass pea.

The manuscript is written clearly, and the methods and results look solid to me. I have checked the supplementary materials and the data described in the paper. I have no suggestions for revision.

Reviewer #2:

Remarks to the Author:

Grass pea, one of the oldest crops in the world, might be the sustainable and resilient answer to climate challenges. The lack of genetic resources and the crop's association with the disease *neurolethyrism* have limited the cultivation of grass pea. Therefore, genome sequencing in this manuscript is very important to this under-utilization crop. Generally, the manuscript provides some new information. However, the authors have plenty for improvement.

1. Firstly, while the authors present a new genomic resource of *Lathyrus sativus*, the genome assembly does not reach the quality of newer assemblies derived from high-accuracy long reads. In addition, no attempt was made to assign the assemble onto chromosomes, and Hi-C, BioNano or genetic map could be useful to improve the assemble significantly.

The assembly and annotation require additional QC and information. The quality of the assembly was evaluated only in terms of sequence stats (e.g. N50) and BUSCO completeness. Now days there are good tools to assess the quality of the assembly in terms of completeness, region collapsing and so on. Merqury (<https://github.com/marbl/merqury>) should be used to assess assembly completeness. The reference genome heterozygosity what it is usually useful in terms of understanding the assembly challenges is missing. Probably the genome size estimation with the K-mer distribution could help to understand the proportion of the genome.

2. There are some methodological omissions. Firstly, the authors do not describe any screening for organelle fragments and contamination. Also, there is no mention of the assembly of the chloroplast or mitochondria genomes.

3. Aside from the genomic resource, there need more novel scientific discoveries on general interests. Using this genome sequence, authors try to elucidate the biosynthetic pathway leading to the formation of β -ODAP. However, all the results are mostly based on the previous research results.

4. Authors defined β -ODAP as neurotoxin. In fact, β -ODAP had been discovered as a neuroactive non-protein amino acid, while cause *neurolethyrism* with prolonged overconsumption of grass pea. Moreover, β -ODAP functions in hemorrhage stopping and thrombopoiesis treatment, etc. So, β -ODAP described as neurotoxic is just an overemphasis on its toxic characteristics; neuroactive is more better. Maybe the published paper in *Planta* (Lambein, et al., (2019) Grass pea (*Lathyrus sativus* L.): orphan crop, nutraceutical or just plain food?) is helpful.

5. What is the difference between neurolathyrism and lathyrism? Authors used neurolathyrism in text but lathyrism in keywords. And similarly β -L-ODAP and β -ODAP.
6. One of the most important results in this manuscript is the ability of LsAAE3 to synthesise β -ODAP and α -ODAP in vitro, and the LsAAE3-LsBOS coupled reaction producing β -ODAP at pH 7.0. Why MtAAE3 was used as control but not PsAAE3 from *Pisum sativum*? L-DAP, the immediate precursor for β -ODAP synthesis, is present in grass pea and pea tissues but not in *M. truncatula*. So, how about the activity of PsAAE3 is more interesting and necessary to clarify the biosynthesis of β -ODAP. The grass pea genome contains a AAE3 gene encoding an enzyme with 75% amino acid identity to AtAAE3 and 88% amino acid identity to MtAAE3. However, LsAAE3 shares 96.35% amino acid identity to PsAAE3, lacking in this manuscript. What is the difference between PsAAE3 and LsAAE3? It is very very important to this manuscript. The authors should apply a transgenic research to provide in vivo support for their claim.
7. Why heterologous expression of LsBOS (BAHD) in *Nicotiana benthamiana* resulted in the formation of β -ODAP but failed when purified LsBOS was incubated with oxalyl-CoA and L-DAP? It should be discussed. Authors named one of (BAHD 2, 3, 8 and 9) as β -L-ODAP synthase (LsBOS), which one?
8. In fig 3a, authors showed the grass pea β -ODAP biosynthesis as posited by Yan et al. 2006. However, Yan et al. 2006 just reviewed the pathway of β -ODAP biosynthesis. The key enzyme of CAS were primarily verified by Xu et al., 2017; Song et al., 2021; and BOS by Emmrich 2017.
9. In fig 3d and 3e, what is the difference between β -ODAP and α , β -ODAP standard? As we all know, purified β -ODAP or β -ODAP standard from sigma contained its isomer α -ODAP. How to get a single peak of β -ODAP in fig 3d, but two peaks in fig 3e and fig S7 with the same detect condition? And, what is the possible mechanism of LsAAE3 to synthesis β -ODAP and α -ODAP but only β -ODAP when coupled with LsBOS?
10. In table S7, gene models were assigned protein ranks based on coverage compared to a database consisting of gene models of nine plant species including even *Cucumis sativus*, *Malus domestica*, *Prunus persica* but with *Pisum sativum* lacking. In fact, *Pisum sativum* is generally used as control in β -ODAP biosynthesis.
11. Authors suggested that the ability of LsAAE3 to synthesise β -ODAP and α -ODAP in the absence of LsBOS in vitro, may explain why no β -ODAP-free mutants of grass pea have yet been identified. In fact, as Professor Lambein described, the link of β -ODAP biosynthesis with primary metabolism (Liu et al., 2017, *J Agr Food Chem* 2017, 65(47): 10206-10213) will explain the difficulty to lower the content of β -ODAP by classical breeding, especially to develop varieties with zero ODAP.
12. Interaction between LsAAE3 and LsBOS is an important supporting for their coupled reaction reported in this manuscript, however, only surface plasmon resonance was used to investigate the LsAAE3-LsBOS interaction. Co-IP, BiFC...etc. is necessary to strengthen the conclusion.
13. The discussion is far too long. I believe it has two major flaws: Firstly, it reads like an introduction in parts. Discussions should not have excess background information. Such background should be provided in the Introduction. Secondly, there is a lot of summarizing in the discussion, which should also be avoided. There is no need to state results twice.
14. The writing still requires additional English grammar editing to address minor problems, especially the style of references.

Reviewer #3:

Remarks to the Author:

Grass pea (*Lathyrus sativus* L.) is a traditional legume crop associated with the disease neurolathyrism. The likely cause is a non-protein amino acid, β -L-oxalyl-2,3-diaminopropionic acid (β -L-ODAP). RNASeq was used to establish the biosynthesis of β -L-ODAP. Genes/enzymes identified are acyl-activating enzyme 3 (LsAAE3) and a novel BAHD-acyltransferase (LsBOS) that form a metabolon activated by CoA to produce β -L-ODAP. This information is important for crop improvement via Tilling or genome editing.

Genome analysis and RNASeq use latest technologies and provided important information of relevant

gens of β -L-ODAP biosynthesis. The authors cloned the respective genes and could demonstrate corresponding enzymatic activities. Thus the biosynthesis was unequivocally established. The ms is very well written, informative and concise. It covers the relevant literature. It is a major advance in our understanding of β -L-ODAP biosynthesis.

Point-by-point responses to reviewers' comments

We are grateful to all three reviewers for taking the time to appraise our work and for their suggestions about how to improve the manuscript. Our point-by-point responses to the reviewer's comments and our changes to the manuscript are detailed below:

Reviewer 1

[...] The manuscript is written clearly, and the methods and results look solid to me. I have checked the supplementary materials and the data described in the paper. I have no suggestions for revision.

Response: We thank reviewer 2 for their positive review. This reviewer highlighted the large size of the grass pea genome and considered the biochemical data presented in the manuscript to be convincing.

Reviewer 2

1. Firstly, while the authors present a new genomic resource of *Lathyrus sativus*, the genome assembly does not reach the quality of newer assemblies derived from high-accuracy long reads. In addition, no attempt was made to assign the assemble onto chromosomes, and Hi-C, BioNano or genetic map could be useful to improve the assemble significantly.

Response: Reviewer 2's suggestion of conducting Hi-C scaffolding was very helpful. We undertook this analysis which allowed us to scaffold part of the assembly into 9 scaffolds, 7 of which are chromosome-scale (results L97ff, methods L781ff). While achieving better macro-resolution, the assembly that incorporated the Hi-C data excluded a number of contigs. Consequently, we have chosen to present the HiC-scaffolds alongside the original polished redbean assembly. Since our analysis of β -L-ODAP synthesis was based on the polished redbean assembly, we believe that providing this draft assembly together with its annotation for use by the community is the best approach, as evidenced by the supportive comments from reviewers 1 and 3. Researchers may choose to use the scaffolded assembly excluding low-quality regions, or the more extensive, but more fragmented, redbean assembly, depending on the goals of their analyses.

2. The assembly and annotation require additional QC and information [...] there is no mention of the assembly of the chloroplast or mitochondria genomes.

Response: To add to the QC of the assembly, we have used BlobToolKit (results L106ff and new supplementary table S3, methods L792ff). This revealed minimal contamination with foreign sequences in the assembly (0.04% proteobacteria, 0.06% chordata, no arthropods, ascomycota, firmicutes or other eukaryotes). We selected out the plastid and mitochondrial genomes and provide these as two additional supplementary datasets.

3. Aside from the genomic resource, there need more novel scientific discoveries on general interests.

Response: We used the genome assembly to identify genes that had already been discovered in the pathway but were able to locate them in a genome assembly and assess copy numbers of these genes.

Following up on our biochemical analyses of genes encoding enzymes of β -L-ODAP biosynthesis we defined the unique mechanism of action of LsAAE3 and LsBOS in synthesising β -L-ODAP. This insight was derived from our analysis of the genome of grass pea and is completely novel. Based on our draft genome sequence, we have enhanced the understanding of β -L-ODAP synthesis by establishing:

a) the numbers of genes encoding each of the steps of β -L-ODAP synthesis that have already been reported. This is important for reverse genetic screening for low/zero β -L-ODAP mutants or targeted gene editing

b) The adenylase activity of LsAAE3 with L-DAP and oxalate to form β -L-ODAP and α -L-ODAP.

c) LsBOS enhances this activity of LsAAE3 significantly and gives stereospecificity for β -L-ODAP.

d) LsAAE3 and LsBOS interact physically, forming a metabolon, the first time such a strong interaction has been shown using SPR for plant metabolons

e) CoA serves as an activator rather than as a substrate in the synthesis of β -L-ODAP.

f) we measured L-DAP for the first time in grass pea accessions as well as in pea, confirming its role as an intermediate in β -L-ODAP synthesis.

4. Authors defined β -ODAP as neurotoxin. In fact, β -ODAP had been discovered as a neuroactive non-protein amino acid, while cause neurolathyrism with prolonged overconsumption of grass pea. Moreover, β -ODAP functions in hemorrhage stopping and thrombopoiesis treatment, etc. So, β -ODAP described as neurotoxic is just an overemphasis on its toxic characteristics; neuroactive is more better.

Response: Several publications have suggested health promoting effects of β -L-ODAP, especially from Panax notoginseng when used in topical applications. However, the contribution of β -L-ODAP contained in grass pea to neurolathyrism epidemics is well established and is the reason why breeding low β -L-ODAP varieties remains the main breeding target of ICARDA (for whom grass pea is a mandate crop), and countries for which grass pea is the major grain legume, such as Bangladesh and Ethiopia. To address the reviewer's point we have changed the manuscript text to introduce β -L-ODAP as a "neuroactive compound" (L53) and added the sentence "Pharmacological and nutraceutical uses of β -L-ODAP have been proposed¹⁶, but its role in the aetiology of neurolathyrism remains the primary limitation on more widespread use of grass pea as a food and feed." (L57ff), including the reference mentioned by the reviewer. We have opted to keep the language of β -L-ODAP as a toxin in the abstract to emphasise the importance of our results for food safety and crop breeding (L65f).

5. What is difference between neurolathyrism and lathyrism? Authors used neurolathyrism in text but lathyrism in keywords. And similarly β -L-ODAP and β -ODAP?

Response: The terms "lathyrism" and "neurolathyrism" are sometimes used interchangeably in the literature. In the manuscript, we consistently refer to "neurolathyrism" to avoid confusion with the distinct syndromes osteolathyrism and angiopathyrism caused by other toxins found in Lathyrus spp. We have corrected the keyword to "neurolathyrism". Since β -D-ODAP does not occur naturally, all literature concerning β -ODAP in grass pea refers to the L-enantiomer. The reviewer is right that this should be specified for the sake of precision, and we have corrected the keyword and any other instances in our manuscript mentioning " β -ODAP" to " β -L-ODAP".

6. One of the most important results in this manuscript is the ability of LsAAE3 to synthesise β -ODAP and α -ODAP in vitro, and the LsAAE3-LsBOS coupled reaction producing β -ODAP at pH 7.0. Why MtAAE3 was used as control but not PsAAE3 from *Pisum sativum*? L-DAP, the immediate precursor for β -ODAP synthesis, is present in grass pea and pea tissues but not in

M. truncatula. So, How about the activity of PsAAE3 is more interesting and necessary to clarify the biosynthesis of β -ODAP.

Response: Perhaps the reviewer is wondering why pea does not make β -L-ODAP if it has an AAE3 protein that is very similar to that in grass pea, and accumulates low levels of the substrate L-DAP? In response to the reviewer's request, we expressed the pea enzyme PsAAE3 and found it to have the same pH optimum as LsAAE3. PsAAE3 can also synthesise β -L-ODAP when combined with LsBOS. The simple explanation for why pea does not produce β -L-ODAP is therefore because it does not produce BOS. The activity of AAE3 from pea or grass pea in producing β -L-ODAP from oxalate and L-DAP is two orders of magnitude lower than when LsBOS is included in the assay in vitro (irrespective of the presence of CoA). Therefore, the ability of PsAAE3 to make β -L-ODAP from oxalate and L-DAP in vitro would not be of physiological relevance, and any β -L-ODAP produced in pea by the activity of PsAAE3 alone would be below our current detection limits.

The authors should apply a transgenic research to provide in vivo support for their claim.

Response: In their reference to "transgenic research", we assume reviewer 2 is suggesting that we try to silence LsAAE3 or LsBOS by transgenic means. Unfortunately, such experiments are technically very challenging in most legumes, including L. sativus, and would add nothing to our understanding of the unique mechanisms of action of these enzymes. Our analyses of the enzyme activities and their interactions in vitro have addressed the mechanisms of action much more comprehensively, demonstrating clearly the value of biochemical analysis for such studies.

7. Why heterologous expression of LsBOS (BAHD) in *Nicotiana benthamiana* resulted in the formation of β -ODAP but failed when purified LsBOS was incubated with oxalyl-CoA and L-DAP? It should be discussed.

Response: The reviewer may have misunderstood the mechanism of action of the LsAAE3-LsBOS metabolon. The adenylase activity is an inherent attribute of all AAE3 enzymes, and when coupled with LsBOS, is able to synthesise β -L-ODAP. This is also possible for NbAAE3 in the presence of L-DAP and LsBOS, and we have added a sentence pointing this out to the discussion (Line 520).

Authors named one of (BAHD 2, 3, 8 and 9) as β -L-ODAP synthase (LsBOS), which one?

Response: We renamed BAHD3 as LsBOS, as shown in Fig. 3c. For clarity, we have amended Line 204 to state this explicitly.

8. In fig 3a, authors showed the grass pea β -ODAP biosynthesis as posited by Yan et al. 2006. However, Yan et al. 2006 just reviewed the pathway of β -ODAP biosynthesis. The key enzyme of CAS were primarily verified by Xu et al., 2017; Song et al., 2021; and BOS by Emmrich 2017

Response: The reviewer is right that individual enzymes were first verified by the authors named, as referenced in the main text. We cited Yan et al. 2006 as they reviewed the entire pathway and their model still forms the consensus in the literature (which our research amends, as shown in Fig. 4c). To clarify the sources of the information presented here, we changed the caption of Fig. 3a to read "grass pea β -L-ODAP biosynthesis as reviewed by Yan et al." instead of "[...] posited by Yan et al." and added citations besides each of the enzymes in Fig. 3a.

9. In fig 3d and 3e, what is the difference between β -ODAP and α,β -ODAP standard?

*Response: The standard in question was obtained from Lathyrus Technologies, not from Sigma, and was indeed a β -L-ODAP standard, free of the α -isomer (it is possible to separate the two isomers chromatographically). In order to also identify α -L-ODAP, we took an aliquot of our β -L-ODAP standard, dissolved it in dH₂O and allowed it to isomerise by repeated freeze-thawing, forming an equilibrium between both isomers, as described by Zhao et al. in *Phys. Chem. Chem. Phys.*, 1999, 1, 3771-3773. This resulted in the distinct β -L-ODAP and α/β -L-ODAP standards shown in Fig. 3 di and ei, respectively. We have added a note to the Materials and Methods section in line 902 to clarify this.*

And, what is the possible mechanism of LsAAE3 to synthesis β -ODAP and α -ODAP but only β -ODAP when coupled with LsBOS?

Response: The mechanism whereby LsBOS confers stereospecificity to the LsBOS-LsAAE3 metabolon is an intriguing question and one that will require structural studies of the entire protein complex including all ligands. To clarify this, we have amended the sentence in the discussion (L380) reading “The presence of LsBOS provided regio-specificity to this reaction to produce exclusively β -L-ODAP” with “, but the catalytic mechanism awaits structural resolution of the complex”.

10. In table S7, gene models were assigned protein ranks based on coverage compared to a database consisting of gene models of nine plant species including even *Cucumis sativus*, *Malus domestica*, *Prunus persica* but with *Pisum sativum* lacking. In fact, *Pisum sativum* is generally used as control in β -ODAP biosynthesis.

*The species we used for the gene model prediction (in what is now table S8) were *Cicer arietinum*, *Cucumis sativus*, *Fragaria vesca*, *Glycine max*, *Malus domestica*, *Medicago truncatula*, *Prunus persica*, *Phaseolus vulgaris*, and *Trifolium pratense* - 5 legume and 4 non-legume species – which we chose for the high quality of their genome annotations to minimise the risk of annotation errors being carried over from one species to another. The pea genome published by Kreplak et al. (2019), with its in-silico annotation relying on gene models from *C. arietinum*, *G. max* and *M. truncatula*, only became available after this part of our work had already been completed. Adding *Pisum sativum* gene models to the gene model prediction would require us to re-run most of the genome annotation pipeline, with questionable gains to the accuracy of our annotation pipeline, which already includes the gene model data used to generate the *P. sativum* annotation along with *L. sativus* RNAseq data.*

11. [...] as Professor Lambein described, the link of β -ODAP biosynthesis with primary metabolism (Liu et al., 2017, *J Agr Food Chem* 2017, 65(47): 10206-10213) will explain the difficulty to lower the content of β -ODAP by classical breeding, especially to develop varieties with zero ODAP.

Response: β -L-ODAP synthesis (just like any other specialised metabolite) is linked to primary metabolism. Both Liu et al. 2017 and the work of Lambein and his co-authors present β -L-ODAP as a final product several steps downstream from primary metabolism. While β -L-ODAP accumulation in grass pea is affected (positively and negatively) by a wide range of environmental factors and metabolic disturbances, this does not necessarily mean that disruption of enzymatic β -L-ODAP synthesis downstream of primary metabolism would disrupt primary metabolism critically, as reviewer 2 implies. The detailed investigation of the link between primary metabolism and β -L-ODAP synthesis will require knockouts of the enzyme-encoding genes we describe here to measure impacts on primary metabolism, as well as whole plant environmental responses.

To address the impact of β -L-ODAP synthesis in planta, we have investigated the effects of oxalate toxicity in a high (LS007) and a low β -L-ODAP variety (Mahateora). Mahateora appears to be more

sensitive to oxalate than LS007 supporting our suggestion that β -L-ODAP synthesis may have evolved as a means of removing oxalate (results L286ff and supplementary Figure S9, methods L981ff)

12. Interaction between LsAAE3 and LsBOS is an important supporting for their coupled reaction reported in this manuscript, however, only surface plasmon resonance was used to investigate the LsAAE3-LsBOS interaction. Co-IP, BiFC...etc. is necessary to strength the conclusion.

Response: We thank the reviewer for suggesting additional techniques to confirm the protein-protein interaction in the LsAAE3-LsBOS metabolon Co-IP and BiFC are standard techniques to establish protein-protein interactions. SPR provides the most robust evidence for interactions in a metabolon, including stoichiometric data. In fact, no plant metabolon has yet been analysed using SPR, as the interactions are typically too weak to be observed with this technique. To address reviewer 2's point, we have measured the interaction of S-tagged LsAAE3 and His-tagged LsBOS using Co-IP, showing reciprocal co-precipitation (results L279ff and new supplementary figure S8, methods L956ff). Intriguingly, co-precipitation was strengthened in the presence of L-DAP, which further supports the importance of the metabolon to the functioning of the LsAAE3-LsBOS coupled activity.

13. The discussion is far too long. The writing still requires additional English grammar editing to address minor problems, especially the style of references.

Response: We have cut down the length of the discussion by ~20% and reviewed the references for uniformity of style. We have checked the English grammar and its use carefully and believe that the manuscript is well written as commented by reviewers 1 and 3.

Reviewer 3

[...] Genome analysis and RNASeq use latest technologies and provided important information of relevant gens of β -L-ODAP biosynthesis. The authors cloned the respective genes and could demonstrate corresponding enzymatic activities. Thus the biosynthesis was unequivocally established. The ms is very well written, informative and concise. It covers the relevant literature. It is a major advance in our understanding of β -L-ODAP biosynthesis.

Response: We thank reviewer 3 for this highly positive review, specifically noting the value of our biochemical work in understanding the β -L-ODAP pathway.

Reviewers' Comments:

Reviewer #2:

I noticed that the quality of the manuscript has been greatly improved by considering the recommendations of the reviewers. I have no further suggestions for revision.